

# Spatial Patterns of Snow Distribution for Improved Earth System Modelling in the Arctic

Katrina E. Bennett[1], Greta Miller[1], Robert Busey[2], Min Chen[1], Emma R. Lathrop[1], Julian B. Dann[1], Mara Nutt[1], Ryan Crumley[1], Baptiste Dafflon[3], Jitendra Kumar[4], W. Robert Bolton[2], and Cathy J. Wilson[1]

[1] Los Alamos National Laboratory, Earth and Environmental Sciences, Los Alamos, NM
[2] University of Alaska Fairbanks, International Arctic Research Center, Fairbanks, AK
[3] Lawrence Berkeley National Laboratory, Berkeley, CA
[4] Oak Ridge National Laboratory, Oak Ridge, TN

*Correspondence to*: Katrina E. Bennett (kbennett@lanl.gov)

**Abstract.** The spatial distribution of snow plays a vital role in Arctic climate, hydrology, and ecology due to its fundamental influence on the water balance, thermal regimes, vegetation, and carbon flux. However, for earth system modelling, the spatial distribution of snow is not well understood, and therefore, it is not

well modeled, which can lead to substantial uncertainties in snow cover representations. To capture key hydro-ecological controls on snow spatial distribution, we carried out intensive field studies over multiple years for two small (2017-2019, ~2.5 km$^2$) sub-Arctic study sites located on the Seward Peninsula of Alaska. Using an intensive suite of field observations (>22,000 data points), we developed simple models of spatial distribution of snow water equivalent (SWE) using factors such as topographic characteristics,

vegetation characteristics based on greenness (normalized different vegetation index, NDVI), and a simple metric for approximating winds. The most successful model was the random forest using both study sites and all years, which was able to accurately capture the complexity and variability of snow characteristics across the sites. Approximately 86% of the SWE distribution could be accounted for, on average, by the random forest model at the study sites. Factors that impacted year-to-year snow distribution included

NDVI, elevation, and a metric to represent coarse microtopography (topographic position index, or TPI), while slope, wind, and fine microtopography factors were less important. The models were used to predict SWE at the locations through the study area and for all years. The characterization of the SWE spatial distribution patterns and the statistical relationships developed between SWE and its impacting factors will be used for the improvement of snow distribution modelling in the Department of Energy's earth system

model, and to improve understanding of hydrology, topography, and vegetation dynamics in the Arctic and sub-Arctic regions of the globe.

Keywords: Snow Distribution, Machine Learning, Spatial Patterns, Topography, Vegetation, Permafrost,
Alaska, Arctic

## 1   Introduction

Covering the land for more than six months each year, snow plays a vital role in the climate, hydrology, and ecosystems of the Arctic and sub-arctic. Snow directly impacts climate through modulation of
atmospheric circulation patterns via the snow-albedo feedback mechanism (Fletcher et al., 2009) and atmospheric moisture budgets through its control on the amount of water available for evaporation (Callaghan et al., 2011). Thus, snow plays a fundamental role in controlling water availability, soil





moisture, and temperature, affecting all components of Arctic ecosystem including vegetation (Evans et al., 1989; Schaefer and Messier, 1995; Scott and Rouse, 1995) animal populations (Forchhammer et al., 2008;

Manning and Garton, 2012), microbial decomposition and carbon flux (Mauritz et al., 2017; Zona et al., 2016). The distribution of snow and timing of its melt is key to understanding how changes in hydrology, soil thermal regimes, and vegetation interact across the Arctic landscape (Jafarov et al., 2018). Snow distribution, however, is a particularly elusive and difficult feature to characterize, leading to challenges in how to quantify snow properties over space and understand how snow changes over time, especially in

remote, under-monitored watersheds of the Arctic and sub-Arctic.

Within the Arctic hydrologic cycle, spring snowmelt is the single most important event contributing to the annual water budget of high latitude watersheds (Ford and Bedford, 1987; Stuefer et al., 2013). The characteristics of spring snowmelt also affect extreme runoff events (Marsh et al., 2008), refill thermokarst lakes (Chen et al., 2014), recharge the groundwater (Kane and Stein, 1983) and generally dominate the

annual hydrograph (Carey and Quinton, 2004). The extent, duration, timing, and magnitude of snowmelt also affects ice-jam flooding occurrence in northern rivers (Prowse and Beltaos, 2002; Beltaos and Prowse, 2009; de Rham et al., 2020).

Due to the dominant control of snow on the water balance during the spring, the end-of-winter snow distribution has received much attention (Homan and Kane, 2015; Kane et al., 1991; Woo and Young,

2004). In particular, capturing snow distribution at the end-of-winter is highly valuable to study insulating effects that have a strong effect on active layer thickness (Atchley et al., 2016), permafrost temperature (Stieglitz et al., 2003), and permafrost distribution (Zhang, 2005; Park et al., 2015; O'Neill and Burn, 2017). Snow distribution further impacts the movement of water between surface and subsurface reservoirs, soil water storage capacity, and subsurface flow (Roach et al., 2011; Frampton et al., 2013; Jafarov et al.,

65   2018).

One of the main characteristics of the Arctic end-of-winter snow distribution is its high spatial heterogeneity (Homan and Kane, 2015; Davison et al., 2006; Hinzman and Kane, 1991; Assini and Young, 2012; Woo and Young, 2004). For example, the unitless coefficient of variation in snow depth along 500 m long transects in northern Finland ranged from 0.13 to 0.36, dependent on the land cover    (Hannula et al.,

2016). For a 1.65 km long and 250 m wide transect in Central Yamal, the end-of-winter snow depth ranged from 0 to 3.15 m, with a coefficient of variation 1.31 (Dvornikov et al., 2015). Within a 240 × 160 km domain located on the North Slope of Alaska, the mean end-of-winter snow water equivalent (SWE) coefficient of variation ranged from 0.19 to 0.32 in a 12-year period across years and sites (Stuefer et al., 2013). Despite high spatial heterogeneity in snow distribution across scales, the spatial pattern of snow

distribution is relatively stable from year-to-year for a given location because of the nature of the factors that affect snow accumulation and distribution patterns, including  topography, vegetation, and predominant wind patterns (Liston, 2004; Kirnbauer and Blöschl, 1994; Sturm and Wagner, 2010; Homan and Kane, 2015; Woo and Young, 2004; König and Sturm, 1998; Rees et al., 2014).

Many studies have been conducted to characterize the spatial pattern of snow distribution and investigate

how different factors affect the spatial distribution (sometimes referred to as snow redistribution, but denoted herein as snow distribution) of snow across a range of scales. However, snow distribution is particularly elusive and difficult to characterize, leading to challenges in how to a) quantify snow properties over space, and b) understand how snow changes over time, especially in remote, under-monitored watersheds of the Arctic and sub-Arctic. And, while it is well known that topography, vegetation, and wind

are common factors driving snow distribution, the conclusions on how important those factors are in controlling snow distribution vary depending on the scale and site-specific characteristics. For example, at micro- (10 to100 m) and meso- (100 to 10,000 m) scales, several studies found that shrub distribution has a dominant control on snow depth (Sturm and Wagner, 2010; Sturm et al., 2001a; Vajda et al., 2006), while other studies found that topographic features had a dominant control on snow distribution, including

curvature (Dvornikov et al., 2015), wind (as related to aspect and slope) (Dvornikov et al., 2015; Vajda et al., 2006; Bruland et al., 2001), or elevation and microtopography (Wainwright et al., 2017; Boike et al.,



2019). At macro-scales (10,000 to 100,000 m), it has been found that snow distribution is largely controlled by vegetation (Essery and Pomeroy, 2004), wind and landscape patterns (Pomeroy et al., 1997), terrain types (Rees et al., 2014; Young et al., 2018), and orographic processes interacting with the regional
topographic slope (Sturm and Wagner, 2010).

There are two main approaches to identify and quantify the influence of different factors on snow distribution: physically-based (dynamical) and statistical (empirical) models (Tarboton et al., 2000; Grünewald et al., 2013). Physically-based dynamical models including SnowTran-3D (Liston and Sturm, 1998; Liston et al., 2007; Hirashima et al., 2004), the distributed blowing snow model (DBSM) (Pomeroy
et al., 2007; Essery and Pomeroy, 2004), and the 3D snowdrift model (Jaedicke and Sandvik, 2002) have been successfully applied to the Arctic. These physically-based models account for both mass and energy exchanges and keep track of state variables related to mass and energy, and allow for detailed representation of different snow processes such as deposition, accumulation, redistribution, sublimation and melting (Tarboton et al., 2000; Grünewald et al., 2013). However, high-quality meteorology, topography,
and vegetation parameterizations are generally required as input, and the computational cost can be high for dynamical models (Liston, 2004; Grünewald et al., 2013).

In contrast, empirically-based statistical models use the relationships between snow depth (or snow water equivalent) and topography, vegetation, and wind to predict the snow distribution. Statistical models have parsimonious model structures, so they are computationally inexpensive and easy to use, but their
drawbacks are that they are site-specific and require substantial data for model calibration (Tarboton et al., 2000). Decision trees (König and Sturm, 1998) and multiple linear or non-linear regression models (Wainwright et al., 2017; Dvornikov et al., 2015) are examples of statistical snow distribution models that have been applied in the Arctic region.

More recently, machine learning approaches have been used to quantify snow distribution using a variety
of different algorithms and remote sensing methods. Broxton et al. (2019) applied artificial neural networks to estimate snow density, which was then combined with aerial lidar snow depth to predict SWE. Revuelto et al. (2020) used random forests to predict lidar snow depth distribution from several topographic predictors. King et al. (2020) used random forests for bias correction of a SWE data assimilation product. Other studies have applied machine learning algorithms using remotely sensed observations as predictors,
including brightness temperature, fractional snow-covered area, or the normalized difference snow index (Liu et al., 2020; Bair et al., 2018). While machine learning approaches have shown to be an effective method for predicting snow distribution, few studies have incorporated vegetation characteristics into the models, and few studies have been validated with intensive field observations of SWE (Anderton et al., 2004).

A particular interest of this paper is to investigate the spatial pattern of snow distribution based on intensive field snow sampling surveys and what factors have control on the snow distribution at the local scale for two study sites in the southwestern and central Seward Peninsula, Alaska. Our main focus is on identifying secondary factors of snow distribution as opposed to the primary variables of temperature and precipitation. This study is part of the Next-Generation Ecosystem Experiments for Arctic (NGEE-Arctic) project, which
aims to advance the predictive power of earth system models (ESMs) through understanding of the structure and function of Arctic terrestrial ecosystems (https://ngee-arctic.ornl.gov/).

Specifically, our goal is to build a statistical model to 1) characterize the spatial pattern of the end-of-winter snow distribution, 2) identify the key factors controlling the spatial distribution, and 3) predict the snow distribution for the local study sites. We expect our analysis will be useful for the validation of the
physically-based permafrost hydrology models such as the Advanced Terrestrial Simulator (ATS), which has been developed at fine spatial scales to understand permafrost dynamics for the region (Atchley et al., 2016, 2015; Painter et al., 2016). Further, the statistical snow distribution model will be used to validate and improved snow redistribution in Department of Energy (DOE)'s Energy Exascale Earth System model (E3SM) land surface model (ELM) and to improve understanding of hydrology, topography, and vegetation
dynamics in the Arctic and sub-Arctic regions of the globe.



This paper is organized as follows: in data and methodology, we introduce the two study sites used for our analysis, describe the field observations of snow, and we outline the site characteristics used as factors in the study, including topography, vegetation, and winds. We present the methodology for each modelling approach. We then present the results, and discussion of findings. We summarize the main conclusions for the study, and our next steps in the research.

## 2 Data and Methodology

### 2.1 Study Sites

The Teller watershed (2.3 km$^2$) and Kougarok study site (2.5 km$^2$) are located in the southwest part of the Seward Peninsula, Alaska (Figure 1). The climate of the Seward Peninsula is characterized by cool continental conditions, typified by long, cold winters and short, cool summers, and high precipitation (Peel et al., 2007). The mean annual air temperature at the nearby Nome Municipal Airport (1980-2018) is -2.3 °C, with a mean January temperature of -14.4 °C and mean July temperature of 11.2 °C. Annual precipitation is 430 mm with 45 % falling as snow (National Centers for Environmental Information, National Oceanic and Atmospheric Administration, https://www.ncdc.noaa.gov/). At the Nome Municipal Airport (National Centers for Environmental Information, National Oceanic and Atmospheric Administration, https://www.ncdc.noaa.gov/) average historical (1981-2010) precipitation falling from October to March is 161 mm, while total annual precipitation (rain and snow) is 425 mm. Snow covers the ground, generally, from approximately October through May, dependent on the year.

The Teller watershed, with elevations ranging from 50 m to 300 m, is underlain by discontinuous permafrost, where near-surface permafrost locations are adjacent to zones with no permafrost or deep permafrost table locations overlain by a perennially thawed layer (i.e., talik) (Jorgenson et al., 2008; Busey et al., 2008; Uhlemann et al., 2021; Léger et al., 2019). Surficial geology is characterized by remnants of comprised of shallow metamorphic schist bedrock and highly modified morainal remnants of alluvium in the lowlands, surrounded by pockets of high-grade metasedimentary and metaigneous rocks and shist (Karlstrom, 1964; Till et al., 2011). The Sinuk River is about 1 km to the south of the study site, and there are a few small streams within this catchment that connect to Sinuk River. Along the small streams, alder-willow shrubs grow, while sedge-willow-dryas tundra and mixed shrub-sedge tussock tundra-bog dominate the rest of the watershed (U.S. Fish and Wildlife Service, 2015; Konduri and Kumar, 2021).

The Kougarok study site is located about 80 km outside of Nome, AK, on the leeward side of the Kigluaik Mountains along a minor drainage of the Kuzitrin River, which flows into the Imuruk Basin and, eventually, into the Bering Sea. Kougarok is situated on a gently rising upland dome that tops out at an elevation of ~110 m. The top of the dome is comprised of metagranitic Late Proterozoic bedrock outcropping while the lower slopes have Quaternary aged sediments composed of peat, alluvial sediment, and interlaced gravel lenses (Hopkins, 1955; Till et al., 2011). The site is overlain by an active soil layer containing organic peat and mineral horizons (McCaully et al., 2021), vegetated by alder shrubland, tussock tundra, alder savanna, and rocky areas dominated by dwarf shrubs and lichens (Iversen et al., 2019; Salmon et al., 2016). The site is underlain by permafrost approximately 15–50 m thick, with average thickness in active layer of 56 cm (Hinzman et al., 2003).

### 2.2 Field Observations of Snow

From March 22$^{nd}$ to March 31$^{st}$, 2017 (Teller), March 26$^{th}$ to April 5$^{th}$, 2018 (Teller, Kougarok), and March 31$^{st}$ to April 7$^{th}$, 2019 (Teller), end-of-winter snow surveys were carried out at the study sites to collect snow depth and snow density to calculate SWE. Data were also collected from Teller in 2016, however, that information is not used in the current study. SWE measures the amount of water contained within the snowpack and characterizes the hydrological and thermal impacts of snow cover better than snow depth




(Jonas et al., 2009; Sturm et al., 2010; Liston and Elder, 2006), which is why we focus on SWE in this study.

We measured snow characteristics in several different ways. Snow depth was captured using a Snow-Hydro™ GPS Snow Depth Probe. When the snowpack was greater than 130 cm in depth, the length of the probe, an avalanche probe was used to manually measure the snow depth. Snow bulk density was measured

with a SWE Coring Tube, also manufactured by Snow-Hydro™ (reported error of -9% to 11%, Dixon and Boon, 2012; Young et al., 2018). Approximately three to five bulk density samples were taken for each sampling location, with an interval of ~200-300 meters between measurements. The snow tube was pressed into the snow until the ground was hit and the depth of the snowpack in cm ($Snow\ Depth$) was recorded. After the tube was removed from the snowpack, the snow in the tube was bagged and weighed in grams

($Snow\ Weight$). The cross-sectional area of the snow coring tube ($Coring\ Tube\ Area$) was 30 cm$^3$. Then $Snow\ Density$ (g/cm$^2$) was calculated from Equation (1):

$$Snow\ Density\ =\ \frac{Snow\ Weight}{Snow\ Depth\ \times\ Coring\ Tube\ Area}\ . \tag{1}$$

Inverse distance weighting (IDW) was used to assign snow density for each of the observed snow depth locations. SWE (cm) was then calculated from the average of the snow bulk density measurements using Equation (2), where $Water\ Density$ was 0.997 g/cm$^3$:

$$SWE = Snow\ Depth\ \times\ \frac{Snow\ Density}{Water\ Density}. \tag{2}$$

SWE was then used as the response variable in our statistical models.

## 2.3   Site Characteristics

To model snow redistribution, various landscape factors were estimated for topographic, vegetation, and wind characteristics, as described below. An overview of data sources, factors, and descriptions of the factors is given in Table 1.

### 2.3.1   Topography

To evaluate the effects of topography on SWE distributions, a digital elevation model (DEM) was analyzed to estimate elevation, aspect, and slope for each of the study locations. The Teller and Kougarok DEMs were derived at a 5 m resolution from Interferometric Synthetic Aperture Radar (IfSAR) data available from http://ifsar.gina.alaska.edu/.

To consider the effects of topography at different spatial scales on SWE distributions, we included a fine-scale "microtopography" and a topographic position index (TPI) derived from the DEM following the method of López-Moreno (2009). Both microtopography and TPI provide information on a DEM cell's position relative to the surrounding relief, but at different spatial scales. The value for each cell is given by the difference between that cell and the average elevation of all cells in a surrounding square window of 15

m width for microtopography, and 155 m width for TPI.

By using both microtopography at the finer 15 m scale and TPI at the coarser 155 m scale in the model, the topographic inputs capture the range of terrain variability across spatial scales. Microtopography in the Arctic represents the hummocks and hills in the landscape that can range, in this landscape, from sub-meter (tussocks), to 1-10 m (hummocks, also called frost boils, formed by frost heaving processes in permafrost)

in scale (Sturm and Holmgren, 1994). The 15 m microtopography in this study is the finest possible microtopography that can be derived using a DEM at 5 m resolution, and is essentially equal to the curvature of the terrain (Jenness, 2006). We also consider coarse-scale topographic features, such as permafrost slumps and the river channel (10s to 100s of m), that we defined using TPI. To define the scale of TPI, we applied a smoothing-average window at a range of widths, and then picked the optimal width

with the best random forest model performance and highest feature importance (Figure A1).



### 2.3.2 Vegetation

Owing to the importance of shrubs for trapping drifting snow in the Arctic (Sturm et al., 2001a; Essery and Pomeroy, 2004; Dvornikov et al., 2015; McFadden et al., 2001), we considered different vegetation indicators to reflect types and distributions: vegetation type and vegetation greenness.

Vegetation type was extracted from an updated vegetation map for both study sites (Konduri and Kumar, 2021). From this map, vegetation type was included in the model as a continuous feature ranked according to IDW-interpolated SWE. The continuous ranking was used so that feature importance could be compared across all model features on a consistent basis. For each year and site, each vegetation type was given a relative ranking ranging from 0-1 based on the average observed SWE for that vegetation type. The relative

rankings from each year and site were then averaged to produce a final ranking used in the models (Table A1). Since not all years and sites had samples for each vegetation type, this method allowed us to create a ranking for all vegetation types across all years and sites.

As high-resolution vegetation distribution information is not widely available for most of the Arctic, Normalized Difference Vegetation Index (NDVI) was used to approximate vegetation characteristics.

NDVI is indicative of the abundance of photosynthetically active vegetation (Rouse et al., 1974) and is useful to capture the branch abundance, deciduous canopy cover and maximum height of shrubs in the Arctic tundra landscape (Boelman et al., 2011). NDVI was derived from the 8-band WorldView-2 images obtained on July 27, 2011 at 1.5m resolution for Teller, and July 14, 2017 for Kougarok. Both images were downloaded from the DigitalGlobe website (https://www.digitalglobe.com/).

### 245 2.3.3 Wind

Previous studies showed that snow is usually accumulated on leeward slopes and blown away from windward slopes (Evans et al., 1989; Liston and Sturm, 1998), so we explored whether the prevailing wind would have an impact on the snow distribution. To understand wind patterns, weather stations within the study locations and from the nearby Nome weather station were analyzed (Table A2).

Because faster wind speeds have a greater impact on the redistribution of snow, we considered winter (October to March) wind speeds greater than 5 m/s (Table A2, Liston and Sturm, 1998; Berg, 1986; Sturm and Wagner, 2010; Sturm and Stuefer, 2013). At the weather station at the top of the Teller watershed, the prevailing wind direction (wind speeds > 5 m/s) was the average of data from the 2016-2017, 2017-2018, and 2018-2019 winters, and at the Kougarok weather station, the prevailing wind direction (wind speeds >

5 m/s) was based on data from the 2018-2019 winter only due to instrument issues in previous years. The prevailing wind direction (wind speeds > 5 m/s) for each study site was used to represent the exposure of a particular location to wind as a function of aspect (Dvornikov et al., 2015). We divided the prevailing winds into the eight cardinal and ordinal directions (N, NE, E, SE, S, SW, W, NW) using 45-degree bins, and then derived a unique wind and aspect factor equation for each directional bin. For Teller, the

prevailing wind direction was 102° (E), so the wind and aspect factor ($W$) was calculated using Equation 3:

$$W = -\sin(A), \qquad (3)$$

where $A$ is the aspect (Dvornikov et al., 2015; Evans et al., 1989; Liston and Sturm, 1998). For Kougarok, the prevailing wind direction was 45° (NE), so the $W$ was calculated using Equation 4:

$$W = -\cos(A) - \sin(A). \qquad (4)$$

The wind and aspect factor gives positive values for leeward slopes and negative values for windward slopes, since snow is known to blow away from windward slopes and accumulate on leeward slopes (Dvornikov et al., 2015).



## 3    Methodology

### 3.1    Modelling

We fit three different types of models, linear regression, general additive, and random forests, to quantify the impacts of different factors on snow distribution and characterize the spatial pattern of the snow distribution.

### 3.1.1 Linear Regression Model

The linear regression model is shown in Equation (5),

$$y = \beta_o + \beta_1 x_1 + \cdots \beta_p x_p + \epsilon, \tag{5}$$

where $y$ denotes snow water equivalent (SWE), and $\beta$ are the coefficient of factors expressed by the weighted sum of its $p$ features with an error term, $\epsilon$. Linear models are useful in that they produce simple linear relationships between the response variable and factors, and the factor rankings that linear models produce are easy to interpret. However, relationships between snow distribution and factors may not be

linear, and thus linear models may not provide a good result, additionally, linear models are susceptible to correlations between parameters. The linear models were implemented using the Linear Regression class of the Scikit-learn package in Python (Pedregosa et al., 2011).

### 3.1.2 Generalized Additive Model

A generalized additive model, or GAM, is a class of statistical models in which the usual linear relationship between the response and predictors are replaced by several nonlinear smooth functions to model and capture the non-linearities in the data, i.e.,

$$g(E_Y(y|x)) = \beta_o + f_1(x_1) + f_2(x_2) + \cdots + f_p(x_p), \tag{6}$$

where $g$ denotes snow water equivalent (SWE) as a link function that links the expected value to the

predictor variables $x_1$ ,… $x_p$, which denote smooth, nonparametric functions. This type of model is useful because it allows for non-linear relationships between SWE and the factors. GAMs are generally easy to interpret, but while they are non-linear, GAMs still require a fit to a distribution or shape. The GAMs were implemented using the LinearGAM class of the pygam package in Python (Servén et al., 2018).

### 3.1.3 Random Forest Model

Random forests are based on decision trees, a series of yes/no questions asked about our data eventually leading to a predicted class (or continuous value in the case of regression, Breiman, 2001). A random forest model is defined by a large number of individual decision trees that operate as an ensemble. Each individual tree in the random forest generates a vote for classes (classification) or mean prediction

(regression) of the individual trees, and the one with the most votes is used for the final prediction. Random forests are useful in that they are not impacted by correlations in the data, generally can protect against over-fitting, and have built-in feature randomization. The drawback of these models is that they can be difficult to control, if used in black box mode, and feature importance can be difficult to interpret in comparison to linear modeling or GAM approaches. The random forests were implemented using the

RandomForestRegressor class of the Scikit-learn package in Python (Pedregosa et al., 2011).

### 3.1.4 Model Implementation





For all three model (linear, GAM, random forest) iterations, SWE, the response variable, was square root transformed. Model inputs were normalized to have a mean of 0 and a standard deviation of 1. We utilized a split sample approach for all models, with a train and test set. Performance metrics for all three statistical models were the coefficient of determination ($R^2$) and root mean squared error (RMSE) on the test set, which were evaluated on the untransformed (squared) model output.

We implemented the random forest model using various subsets of the input factors. One implementation, labeled the "final model", was trained on data from all years at Teller and Kougarok combined, with the year included as a feature in the model. We also trained the model separately for only Teller using all years combined (2017-2019), with the year included as a feature in the model. In addition, we implemented individual random forest model runs for each year and each site, without year included as a feature in the model. The model runs on individual sites and years were also implemented for linear regression models and GAMs so that the performance of the three different statistical models could be compared.

Since the random forest performed the best of the three models and has the most comprehensive feature importance metrics, all testing for model features and hyperparameters was completed with the random forest model. The features and hyperparameters were then applied in the linear model and GAM predictions, where applicable. Model hyperparameters (tree density, max depth, max features, min samples split, min samples leaf, and train/test size) were selected using 8-fold cross validation with 200 random combinations of hyperparameters. The selected hyperparameters were further refined to reduce overfitting of the model without affecting model performance on the test set.

Using the random forest model, we measured the value of each input feature in predicting SWE distribution with both impurity feature importance and permutation feature importance. Impurity importance, also known as Mean Decrease in Impurity (MDI) or Gini importance, is proportional to the total number of splits that each feature divides across all trees in the random forest, where features with more splits are more important (Breiman, 2001). While impurity importance is computationally cheap, it is biased towards features with many possible split points, and suffers from overfitting to the training set. Permutation importance, also known as Mean Decrease in Accuracy (MDA), is based on the decrease in model performance when a single feature is randomly shuffled, and more important features result in larger decreases in performance when permuted (Breiman, 2001). Permutation importance is more computationally expensive. We measured permutation importance on the test set. For this study, we analyzed both importance metrics to ensure that the relative feature importance rankings generally agreed.

Due to the decorrelation effects involved in bootstrapping, the predictive accuracy of random forests is generally robust to collinearity across features (Dormann et al., 2013). However, feature collinearity can still be an issue for determining feature importance (Gregorutti et al., 2017). Prior to modelling, we used variance inflation factors as well as pairwise correlation coefficients to assess collinearity among features and ensure collinearity was not a significant issue.

## 4   Results

### 4.1   Meteorological conditions

Meteorological stations located in the study sites (Figure 1) recorded temperatures of -7.1°C (Teller, 2017-2019, including the top and bottom meteorological stations) / -8.1°C (Kougarok, 2018) during the winter months of October to March, while the Nome airport reported average 2017-2019 winter temperatures of -8.0°C. From 2017-2019, the Nome Airport reported winter temperatures that were slightly warmer than the recent (1981-2010) climatological period (-10.5°C). Precipitation recorded at the Nome Airport climate station indicated that from October to March, precipitation was 13.3 cm, 28.0 cm, and 25.0 cm, while precipitation from December to March at the Nome airport was 7.2 cm, 15.2 cm, and 16.4 cm for 2017, 2018, and 2019, respectively. From October to March, the prevailing wind speed and direction for Teller



(top station) were predominantly East-Northeast (2017-2019), while Kougarok was predominantly Northeast in 2019 (Table A2, Busey et al., 2017). The Kougarok meteorological station experienced issues with its wind sensor in 2018, thus those values are not reported herein. The wind directions experienced at the study sites are similar to that experienced at the Nome Airport (East-Northeast). Wind speeds ranged from 4.9 to 6.7 m/s, much lower than wind speeds reported at the Nome (10.1 to 12.4 m/s) in 2017-2019 (Table A2).

## 4.2 Snow depth, density and SWE

Snow depth was collected at thousands of locations (> 22,000 points) while snow bulk density and SWE were measured at hundreds of locations (Table 2, Figure 2). The snow depth and density surveys were conducted at a relatively small sampling spacing, 1-15 m for snow depth and 200-300 m for snow density, resulting in an extensive dataset that gave a reliable and representative estimate of the variation in snow depth and density for the study sites. The number of observations varied from year to year, with the greatest number of snow depth observations being collected in 2017, followed by 2019, and with 2018 having a similar number of observations to 2018 in Teller. In 2017, the survey was not planned for Kougarok, and in 2019, weather concerns prohibited safe travel to the study site. Note that while Teller's 2017 survey collected the most points, the 2019 survey was the most spatially extensive (not shown). No snowfall occurred during the any of the 2017-2019 end-of-winter snow surveys.

The average snow depth in Teller was observed to be lowest in 2017, and similar depths were noted in 2018 and 2019 (109.0 cm, 106.7 cm, respectively). The average snow depth in Kougarok in 2018 was 75.2 cm. SWE was estimated to be lowest in 2017, lower in Kougarok versus Teller in 2018, and the highest SWE values were recorded in 2019 for Teller (Table 2).

High spatial variability in snow depth was measured for 2017, 2018 (Table 2, Figure 2). Snow depth ranged from 2 to 275 cm (mean 86 cm, standard deviation (SD) 37 cm, coefficient of variation (CV) 0.43). Snow density ranged from 0.116 to 0.451 g cm$^{-3}$ (mean density of 0.288 g cm$^{-3}$). SWE was strongly correlated with snow depth, with correlation coefficients ranging from 0.95 to 0.97 (Figure 3a). Snow density positively correlated with snow depth with correlation coefficients from 0.51 to 0.59 (Figure 3b). SWE was calculated from snow depth and snow density as described previously. SWE ranged from 0.4 to 86.7 cm (with mean 25.4 cm, SD 12.4 cm).

## 4.3 Topographic, Vegetation, and Wind Features

Topographic features in the study sites illustrate the variation across the landscape (Figure 4a and Figure 4b). Elevational gradients are strongest at the Teller watershed, topping out at 300 m, while Kougarok's dome-like feature is approximately 100 m. Slopes in Teller are steeper in the middle of the basin and along the stream banks, while Kougarok's slopes are shallower on the west and steeper to the east, with overall more gentle elevational gradients than Teller. TPI illustrate dominant features such as the solifluction lobes, or terrances as they are sometimes referred to, the stream bank in Teller, and the top of the dome in Kougarok.

Wind and aspect factors illustrate that Teller's aspects are largely unidirectional (south, south east facing), while Kougarok's west hillslope is predominantly S facing, with a north facing slope on the east side over the crest of the dome. NDVI patterns are reflective of shrubs and other various tall and dense shrub patches located across the study sites. Vegetation maps illustrate the differences in the two study sites. Teller contains low-to-mid-slope willow-birch and willow shrub complexes, with Ericaceous dwarf shrub tundra and wet meadows located in the upper slopes. Kougarok's slopes are largely mixed shrub-sedge tussock tundra, with patches of Alder-willow shrubs, dryas-lichen dwarf shrub tundra, Birch-Ericaceous-lichen shrub tundra, and Willow-Birch shrub on the east slope where no snow measurements were taken in 2018 (Konduri and Kumar, 2021).





## 4.4 Modelling

The random forest model hyperparameters were selected using 8-fold cross validation with 200 random
combinations of hyperparameters, and then further refined to reduce overfitting of the model without
affecting model performance on the test set. For the final random forest model, we determined the optimal
train and test split was 80 % and 20 %, the optimal number of trees in the random forest was 300, and the
optimal tree depth was 25. A complete list of the hyperparameters used for the final model and the separate
Teller and Kougarok model runs are given in Table A3.

Several different tests were undertaken to determine the optimal factors used in the final model. We tested
the random forest model to determine the optimal scale of TPI, with a smoothing-average window from 55
m to 505 m (Figure A1). The TPI at a scale of 155 m corresponded to the best model performance. Further,
the random forest model was used to determine which vegetation features to use in the final model. We
tested the model using NDVI, vegetation type as twelve one-hot-encoded categorical features, and
vegetation type as a continuous feature ranked by observed SWE, as well as combinations of these three
features (Figure A2). We found that NDVI and the continuous ranking of vegetation type performed the
best, so these two features were included in the final model. However, while the spatial pattern of
vegetation type improved model performance, the relative order of the vegetation type ranking did not
appear to be important, since other randomized rankings performed similarly well (Figure A2).

The variance inflation factors of the input features are shown in Figure 5. Since the variance inflation
factors are all well below the accepted threshold of 5 (Karimi et al., 2019) and the largest correlation
coefficient between two input variables (0.49 for microtopography and TPI) is well below the accepted
threshold of 0.70, collinearity is not expected to severely distort model estimation and predictions
(Dormann et al., 2013).

The model results for the linear regression, GAM, and random forest for the individual sites and years are
shown in Table 3. To reiterate, we used random forest to select which combination of features were most
successful to predict SWE, and then ran the separate model types using these features for all years
independently. In general, linear regression performed the worst ($R^2$ ranging from 0.31 to 0.44), random
forest performed the best ($R^2$ ranging from 0.72 to 0.92), and GAM performed in between the other two
models ($R^2$ ranging from 0.45 to 0.72). The spatial maps of predicted SWE for the three different models
are shown in Figure A3 for Teller and Figure A4 for Kougarok. Because of the success in simulating SWE
for the study sites and years using random forest, we focus the remainder of our study on the random forest
results to discuss the implications and the driving factors that are ranked to be most important for prediction
of SWE in the study sites.

## 430  4.5 SWE Prediction

The random forest model results for training and testing data are given in Table 4. The final random forest
model which includes data from all years and sites captures approximately 86 % of the variance in SWE
and has an RMSE of 5.85 cm on the test set. The scatter plot of predicted and actual SWE measurements of
the test set from the final model (Figure 6) shows a linear trend. In comparison to the y = x line, the linear
fit in Figure 6 shows the model slightly overestimates low SWE measurements and underestimates high
SWE measurements, which could be due to the tendency of random forests to decrease the variance by
averaging across many trees.

We also considered the random forest model developed using the individual study sites and years as a
feature to predict SWE. The iterations of the random forest model where we considered only Teller and all
years performed slightly worse than the final model, with an $R^2$ of 0.83 and an RMSE of 6.25 cm for Teller
(Table 4). The random forest models trained on individual years and sites ranged in model performance
from $R^2 = 0.72$ and RMSE = 8.51 cm (Teller, 2019) to $R^2 = 0.92$ and RMSE = 5.20 cm (Kougarok, 2018),
as shown in Table 4.



In Figure 7, we illustrate the spatially predicted SWE from the final random forest model for Teller (2017-2019) and Kougarok (2018). SWE values across the basin reflect the year-to-year variability in the amount of precipitation that fell on the study sites. However, we observe that SWE is variable across Teller and Kougarok around major landscape features, such as the stream draws and on the lee side of the permafrost slump features, inside and adjacent to the Teller watershed. Another location where SWE appears to be higher is just below the dome on the west hillslope in Kougarok. The SWE patterns in SWE also reflect areas of higher NDVI values, where shrubs are identified as darker patches in both Teller and Kougarok (see Figure 4).

The spatial error, calculated as the predicted minus the observed for the upper and lower quartiles of error in the random forest SWE prediction is shown in Figure A5 for Teller for each year, and Figure A6 for Kougarok for 2018. In these figures we observe that spatial error varies from year to year, but is not spatially systematic. Errors appear to be higher in the years where there was higher SWE in the basin, such as in 2018 and 2019 compared to the lower SWE year of 2017 in Teller.

### 4.6 Feature Importance

Figure 8 shows the impurity and permutation feature importance results for the final random forest model. Both of the importance metrics provide similar results, and the same variable ranking with the exception of permutation importance metric ranking elevation as higher than NDVI, and impurity ranking NDVI higher than elevation. Overall, NDVI, elevation, and TPI are the most important features for predicting SWE distribution at our study sites. Even though year is ranked as most important feature, this represents the primary drivers of precipitation and temperature that vary from year to year (Bair et al. 2018) and thus is not considered meaningful within the objectives of our study. Features such as slope, vegetation type, wind/aspect factor, and microtopography are ranked as the least important in SWE prediction.

### 4.7 SWE Correlations between Years

Heatmaps that illustrate significant SWE correlations between years 2017-2019 for Teller are shown in Figure 9, based on random forest modeled SWE trained with data from all years and sites. The weakest correlations are for the years with low (2017) and high (2019) SWE years, with stronger correlations between the two higher SWE years (2018 and 2019). These correlations tend highlight the consistency in SWE values for the study site across highly variable climate conditions.

## 5 Discussion

Changes in snowpack characteristics have important implications for a changing Arctic and are anticipated to be a major driver of ecosystem shifts (Bjerke et al., 2015; Cooper, 2014), water and energy balances (AMAP, 2019; Pulliainen et al., 2020), and biodiversity changes (Niittynen et al., 2018; Riseth et al., 2011). Changes in snow have implications for the society of Arctic communities where snow may impact many resources (Huntington et al., 2004), and for global climate change (Overland et al., 2019) and carbon cycles (Rogers et al., 2011; Arndt et al., 2020). Thus, understanding how to better model snow distribution and the important features involved in snow distribution is fundamental to improving how we interpret, and plan for, changing Arctic snow in the future (Zhu et al., 2021; Kouki et al., 2021; Mudryk et al., 2020).

### 5.1 Snow Depth, SWE, and Density Observations

Snow depth and snow density observations collected from two small study sites located on the Seward Peninsula of Alaska comprise an extensive dataset that provides a reliable and representative estimate of the variation in SWE in this region. Snow depth showed high variability in both study sites, with average (all years, sites) coefficients of variation of 0.31-0.57 (Table 2), which was a medium level of variability



compared to the variability range (coefficient of variation of 0.13-1.31) reported for other Arctic regions (Bruland, Sand, and Killingtveit 2001; Hannula et al. 2016; Dvornikov et al. 2015; Stuefer, Kane, and Liston 2013; Homan and Kane 2015; Sturm et al. 2010). These studies note the importance of distance to the Arctic Ocean, which tends to control snow distributions with shallower and less variable snow
thickness near the ocean compared to sites located at a distance from the Arctic Ocean (within upland regions). Compared to snow depth, snow density showed relatively low variability with coefficients of variation of 0.09-0.24 (Table 2) and ranged from 0.12 to 0.62 g/cm$^3$, which was similar to the findings for other non-forest Arctic areas by Homan and Kane (2015) and Hannula et al. (2016) (coefficient of variation of 0.11-0.70).

Consistent with previous studies (Homan and Kane 2015; Assini and Young 2012; Dvornikov et al. 2015; Sturm et al. 2010), there was a high correlation between snow depth and SWE in our study, confirming that SWE is more closely linked to snow depth than to snow density. Sturm et al. (2010) suggested a nonlinear relationship between snow depth and density for a large region of the Northern Hemisphere, while Homan and Kane (2016) did not find any relationship between snow depth and density for a 200 by 240 km region
of Alaska's Central Arctic Slope.

Our study showed an overall positive linear correlation between snow depth and density, indicating that snow depth has some control on the snow density for this catchment. However, snow depth and density showed no relationship for shallow snow (<60 cm), while the linear relationship for deeper snow (>60 cm) was strongest at most sites and years (with the exception of 2018 at Kougarok), which is consistent with
what has been found for a study region consisting of a variety of landscapes in Saskatchewan, Canada (Shook, 1997). The nonlinear relationship between snow depth and density found in the previous work may be due to the fact that the snow surveys documented were conducted for different climate and landscape classes where snow density was largely controlled by climate (wind and temperature) and landscape classes (such as taiga, tundra, mountain, coast). Whereas, in our work, a linear relationship between snow depth
and density was observed in this study because the climate and landscape in the small catchments is more homogenous.

Finally, when we considered SWE between three study years at the Teller site, we found considerable correlations across the years. This is consistent with other research on snow repeat patterns (Sturm and Wagner, 2010; Liston, 2004; Kirnbauer and Blöschl, 1994; Homan and Kane, 2015; Woo and Young,
2004; König and Sturm, 1998; Rees et al., 2014), indicating that there are driving factors that influence snow distribution and those factors are consistent across the year-to-year snow variability (i.e., high and lows).

## 5.2  SWE Modelling and Prediction

The ability to model and predict SWE for our study sites is important on a number of fronts. First of all, we intend to utilize these findings to compare with physics-based modeling efforts, as well as for future machine learning efforts. Our models are being used for investigation of sub-grid SWE variability in E3SM's ELM (Caldwell et al., 2019; Bisht et al., 2018), along with our investigation into ecosystem-type approaches for upscaling of SWE.

The patterns of our SWE maps illustrate the power of utilizing random forest tools over linear methods of estimating SWE distributions (e.g., Broxton et al, 2019, Revuelto et al. 2020, King et al. 2020). When compared to linear and GAM models, we found that random forests significantly outperformed those models. This is mostly likely due to the fact that SWE distributions are controlled by highly non-linear interactions between topography and vegetation characteristics, thus the flexibility offered by the random
forest model can more accurately account for these interactions. While GAM models have been applied successfully to estimate non-linear relationships in snow depth (López-Moreno and Nogués-Bravo, 2005) in the Spanish Pyrenees, they were not as successful at predicting SWE in our study compared to the



random forests model. Random forest also allowed us to test different hypotheses of configurations for the model, determining clearly the success of those configuration and features combinations.

We used multiple study sites with varying characteristics and varying climate years to develop our snow distribution estimates. Because of this unique data set, we were able to develop robust machine learning-based models that we hypothesize are representative of broader SWE patterns across time and space. However, these theories require testing to prove. This hypothesis testing will be incorporated into current and future work that will be carried out using broader observerations of SWE and snow depths to be

collected in an upcoming snow survey planned for 2022 and to be compared with other remote sensed SWE data products.

### 5.3 Features Impacting Snow Distribution

To determine which factors had the most importance to our random forest models, we used two different techniques that were available to us via the random forest modeling, impurity importance and permutation

importance. The two importance measurements generally gave us similar results, which were that the greatest controls on SWE were year (i.e., climate variability), followed by the key secondary factors NDVI, elevation, and TPI. These results in general were consistent with those in previous studies in terms of how those factors affected snow distribution in the Arctic, i.e., more snow was accumulated in the areas with tall shrubs, at higher elevations, and within dominant landscape features, such as within the stream bed and

permafrost thaw slump edges.

NDVI in our study likely reflected the taller, denser shrubs patterns present in the landscape. Although vegetation type in our work was considered less important in our modeling overall, we consider that vegetation within our model was best represented by the signature and patterning on the landscape of the vegetation, which was best represented in the NDVI, rather than the specific type of vegetation (Sturm and

Wagner, 2010). Vegetation type has been noted to play a primary role in end-of-winter snow depth patterning and is also strongly related to variability in winter and spring soil temperatures in the Arctic (Grünberg et al., 2020). The relationships between vegetation type classes and NDVI in our study is being examined as a component of ongoing work for the project.

Our results showed that elevation effects are a dominant factor driving snow distributions at our study sites.

In our study, because random forests can capture both negative and positive relationships, we observe an increase in SWE at higher elevations at Teller due to a slight orographic effect, consistent with the study for a small high-arctic glacier Svenbreen (Małecki 2015), but also likely due to wind blowing into the catchment at the upper wetland meadow. Homan and Kane (2015) discussed a relationship between snow and elevation below specific elevation bands, above which snow is controlled by moisture availability,

however our Teller study site (300 m) is above this elevation threshold, suggesting that threshold may change due to other local or regional factors. However, we observed a decrease in SWE at the top of Kougarok, where snow is removed completely from the upper windswept top (Assini and Young 2012; Shook and Gray 1996; Homan and Kane 2015). For the next phase of our work, we are considering more advanced wind functions (Winstral et al., 2002), and implementing a physically-based wind model for

comparison and testing against our statistical models (Crumley et al., 2021; Liston, 2004).

TPI in our model was found to be the third most important variable, indicating the importance of coarse-scale features in the sub-Arctic landscapes of the Seward Peninsula. These coarse-scale features including stream banks and solifluction lobes/terrace features such as the riser edges are areas of topographic variability where shrub grow and snow accumulates. Thus, they act as hydrology focal points in the basins

where higher enhanced soil moistures and soil warming, and associated increased ecological productivity, can occur. These are also features that act to entrain snow distributed by wind. Indeed, recent research into snowdrift landscape patterns in the Arctic have found that wind transports snow into course-scale features called drift traps, including stream beds, lake features, outcrop features and more. These drift traps contain as much as 40% of SWE found on the landscape and play a significant role in the distribution of snow in



the Arctic (Parr et al., 2020). The relationship between TPI, wind, and repeated snowdrift patterning is under ongoing investigation and the focus of future work.

In our study, microtopography was found to be one of the least important factors driving snow. However, microtopographic patterns were highly correlated with curvature. In research from Dvornikov et al. (2015), curvature was found to have a dominant control on the snow depth at a shrub tundra area in Central Yamal

and there was a positive correlation between shrub heights and snow depth was observed for convex slopes. However, research in fine-scale polygonal tundra sites of the high Arctic, microtopography was found to be an important control on snow (Wainwright et al. 2017). Our study utilized DEMs with a 5 m resolution, with the caveat that microtopography features dominant in this landscape (e.g., drainage paths, terraces) range from centimeters to meters in scale. Thus, we require finer scale DEM sources to investigate this in

more detail, which is the focus of current work based on UAS measurements of snow depths (López-Moreno et al., 2009) and upcoming aerial snow surveys planned for our Alaskan study sites for the end-of-winter 2022.

## 6    Conclusion

The extensive snow depth and density dataset from this study is of high value for calibrating and validating

physically-based models of snow distribution, which is being undertaken in current work by the authors. As the patterns of snow distribution for a given location are similar from year to year, the spatial patterns of snow distribution characterized in this study can be used to represent the typical patterns of snow distribution and model the relative spatial patterns of snow distribution for other years for the study sites. Additionally, this model may be used to estimate snow distribution beyond the study sites, work that is also

ongoing by the authors. The snow distribution pattern and its relationships with the impacting factors found in this study can also be applied to validate and improve the end-of-winter snow distribution at the sub-grid scale in ESMs, which will be undertaken for the DOE's E3SM land surface model, ELM.

We found that random forest models could simulate the SWE distribution most accurately when compared to linear and GAM model approaches, and we were able to simulate the distribution of SWE across the

landscape of these small sub-Arctic study sites. The results of the statistical model are useful for understanding the surface water hydrology during spring snowmelt and explaining differences in permafrost distribution and active layer depth, which have great impact on groundwater hydrology.

Using the random forest, we were able to determine which factors were most important in these sub-Arctic study sites for SWE distribution. These factors were NDVI, that we believe represents shrub patterning,

followed by two topography indexes, elevation, and TPI, an index that represented the features in the landscape such as the stream bed and various topographic features including solifluction lobes. We are currently investigating the relationship between TPI and wind factors, using a more advanced wind estimations and a physically- based wind model. Because of the strong feedback loop between snow, shrubs, and climate (Boike et al., 2019; Sturm et al., 2001a, b), the importance of shrub presence to

estimate SWE highlights the necessity of simulating shrubs at the sub-grid scale in ESMs.

## 7    Data availability

Snow observations collected for this study are available on a publicly available repository, the NGEE Arctic portal (Wilson et al., 2020b; Bennett et al., 2020; Wilson et al., 2020a). Data used as inputs to simulate SWE distribution is available (ifSAR, http://ifsar.gina.alaska.edu/), while the vegetation data are

also available on the NGEE Arctic portal (Konduri and Kumar, 2021). Python modeling codes for developing linear, GAM, and random forest models will be posted on the NGEE Arctic portal.



## 8 Author contribution

KEB conceptualized the random forest modeling, wrote original text, and is the PI of the NGEE Arctic project in 2021; GM completed all modeling and analysis, and figure development; RB was instrumental in field logistics, and observation data collection; MC worked on early modeling iterations, literature review, and original writing; ERL, JBD, and MN provided data collection, and data management; RC edited and contributed to composition; BD edited composition, WRB provided logistical support and observational data collection, along with project management of all collaborative efforts between LANL and UAF; and CJW edited, designed and implemented overall project logistics for field survey, conceptualized earlier implementations of the study design for the composition, oversaw all work, and was PI of the project during the implementation of this work.

## 9 Competing interests

The authors declare that they have no conflict of interest.

## 10 Acknowledgements

Funding for this research was provided by the Department of Energy Office of Science, Office of Biological and Environmental Research through Next Generation Ecosystem Experiment (NGEE)-Arctic project. The authors gratefully acknowledge the contributions of Haruko Wainwright from Berkeley National Lab in earlier versions of this work.



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



Figures

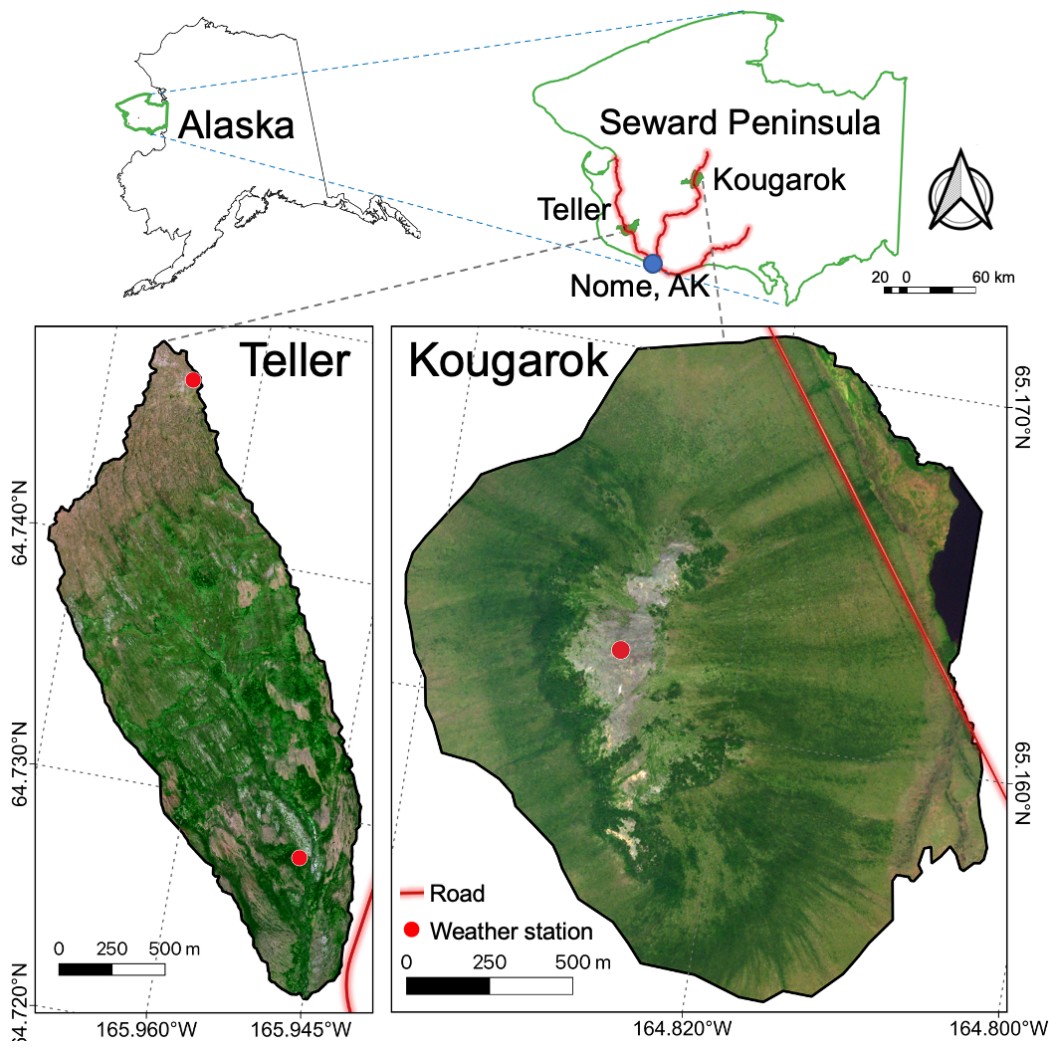

**Figure 1.  Location and WorldView-2 RGB imagery of the Teller watershed (lower left) and Kougarok study site (lower right). Teller has two weather stations (red dots), located near the top and bottom of the watershed, and Kougarok has one weather station located near the top of the rocky dome. The sites are located on the Seward Peninsula of Alaska, with the HUC12 basins for Teller and Kougarok shown in green and the Seward Peninsula road system in red (upper right). RGB composite from the 8-band WorldView-2 images obtained on July 27, 2011 (Teller) and July 14, 2017 (Kougarok) at 1.5m resolution downloaded from the DigitalGlobe website**
**([https://www.digitalglobe.com/](https://www.digitalglobe.com/)).**





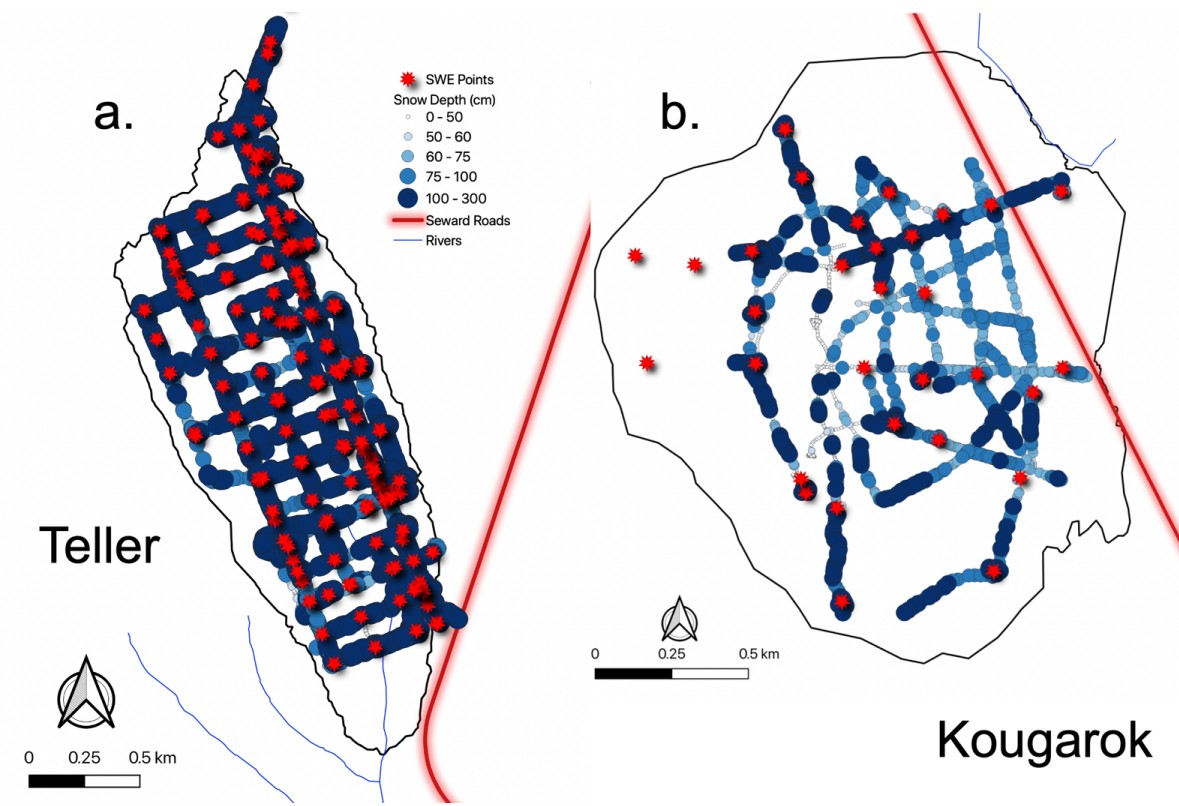

**Figure 2. Measured snow depth (cm) and snow density (SWE Points) at a.) Teller (2017-2019) and b.) Kougarok (2018).**






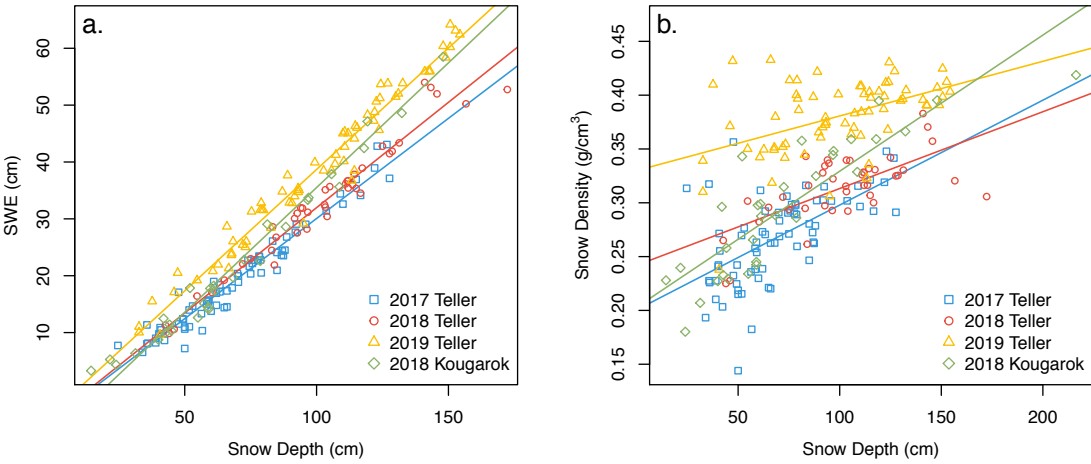

**Figure 3. Scatter plots of (a.) snow depth vs SWE and (b.) snow depth vs snow density. SWE is a function of both snow depth and snow density (see Equation 2).**



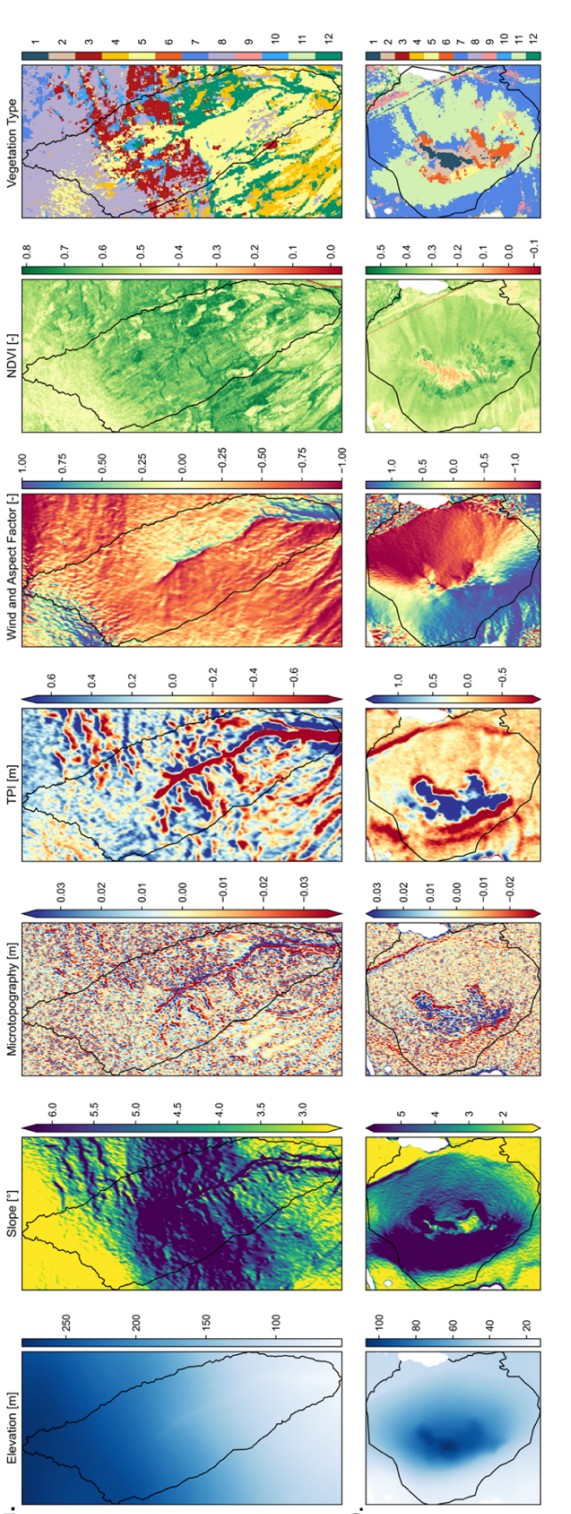

**Figure 4. Input features included in the snow models (elevation, slope, microtopography, TPI, wind and aspect factor, NDVI and vegetation type) for (a.) Teller and (b.) Kougarok. The white areas shown on the Kougarok maps are lakes. The vegetation types are (1) Dryas-lichen dwarf shrub tundra, (2) Birch-Ericaceous-lichen shrub tundra, (3) Ericaceous dwarf shrub tundra, (4) Sedge-willow-Dryas tundra, (5) Willow-birch shrub, (6) Alder–willow shrub, (7) Tussock-lichen tundra, (8) Wet meadow tundra, (9) Wet sedge bog-meadow, (10) Mesic graminoid-herb meadow tundra, (11) Mixed shrub-sedge tussock tundra, and (12) Willow shrub.**




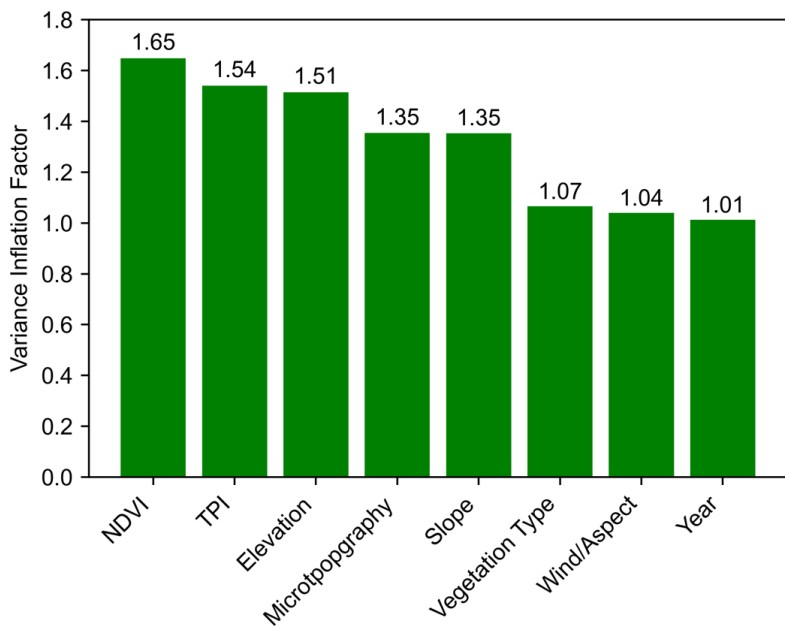

**Figure 5. Variance inflation factors for the model inputs.**







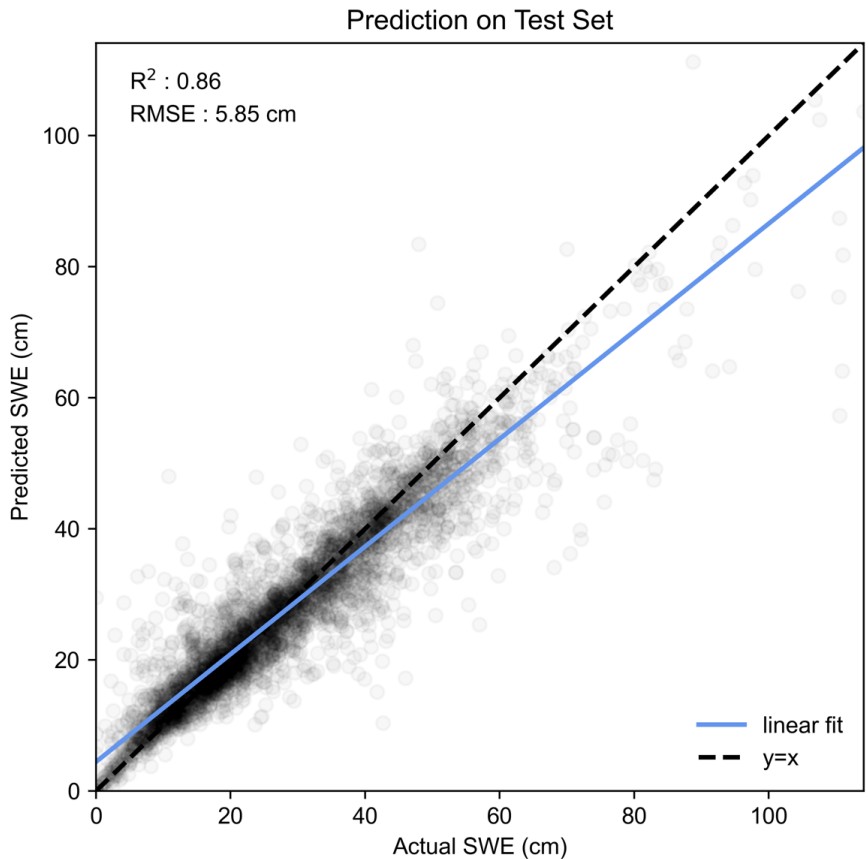

**Figure 6. Random forest results of predicted SWE vs actual SWE for the test set when using all year and site datasets, with a test set of 20%. R$^2$ equal to 0.86 and the RMSE is equal to 5.85 cm. The solid blue line is a linear fit to the scatter points, and the dashed black line is a y=x line.**






**Figure 7.** Spatially predicted SWE for the final model for (a.) Teller 2017, 2018, and 2019 and (b.) Kougarok 2018. The
gray areas on the Kougarok map are small lakes. Note that the scales change for each year and study location across the
panels.





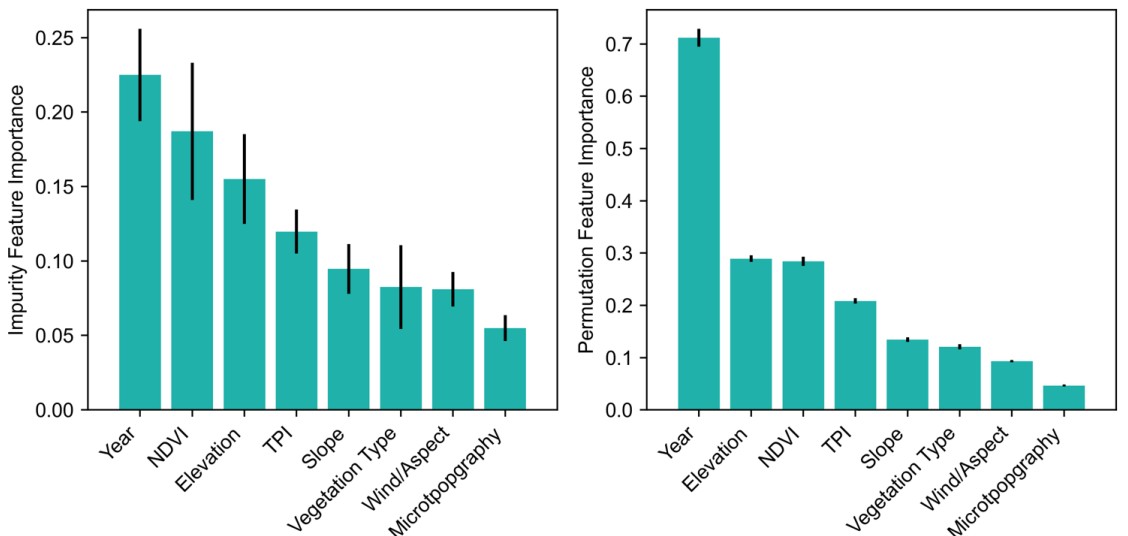

**Figure 8. Random forest feature importance results (left, impurity and right, permutation) for the final model, with ± 1**
**standard deviation shown by the black error bars.**




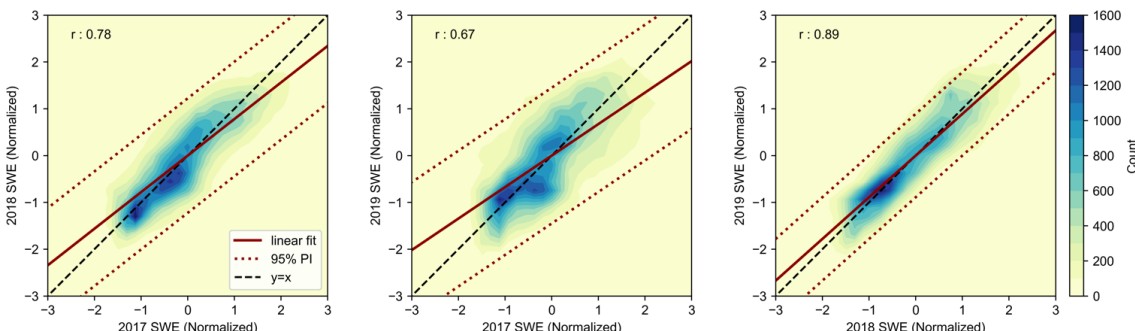


**Figure 9. Heatmaps of predicted normalized SWE for Teller between 2017, 2018 and 2019, showing significant correlations of SWE among years. Random forest was trained with data from all years and sites. The color scale represents the density of data points, with dark blue representing areas with the most points and light yellow representing areas with the least points. The solid red line is a linear fit to the scatter points, the dotted red lines border the 95% prediction interval, and the dashed black line is a y=x line.**




Tables

**Table 1. Topography and vegetation data sources. Data sources for the input features are listed on the left, and the description of the features are listed on the right.**

| Data Sources | Features | Descriptions |
|---|---|---|
| **DEM** | **Elevation** | Elevation (meters) |
| | **Slope** | Slope angle (degrees) |
| | **Microtopography** | Difference between elevation of single 5 m cell and the average elevation of cells in the surrounding box of 15 m width (m) |
| | **Topographic Position Index (TPI)** | Difference between elevation of single 5 m cell and the average elevation of cells in the surrounding box of 155 m width (m) |
| **DEM and wind data** | **Wind/Aspect** | Calculated using wind direction and aspect formula (unitless) |
| **WorldView-2 imagery** | **NDVI** | Normalized difference vegetation index (unitless) |
| **Konduri et al., 2021** | **Vegetation Type** | Vegetation types: |
| | | (1) Dryas-lichen dwarf shrub tundra, (2) Birch-Ericaceous-lichen shrub tundra, (3) Ericaceous dwarf shrub tundra, (4) Sedge-willow-Dryas tundra, (5) Willow-birch shrub, (6) Alder-willow shrub, (7) Tussock-lichen tundra, (8) Wet meadow tundra, (9) Wet sedge bog-meadow, (10) Mesic graminoid-herb meadow tundra, (11) Mixed shrub-sedge tussock tundra, and (12) Willow shrub |

**Table 2. Teller and Kougarok snow depth and density observations and SWE observations and estimates. The coefficient of variation (CV) of the observed variables are in brackets following the average value.**

| Year | | Study Site | Observations | SWE | Density | Snow Depth | SWE |
|---|---|---|---|---|---|---|---|
| | | | Snow depth/SWE (SWE groupings) | Observed, cm (CV) | Observed, g/cm$^3$ (CV) | Observed, cm (CV) | Estimated, cm |
| | 2017 | Teller | 8469 / 234 (77) | 19.33 (0.46) | 0.27 (0.15) | 73.77 (0.42) | 19.37 |
| | 2018 | Teller | 5076 / 150 (49) | 32.57 (0.37) | 0.31 (0.24) | 108.96 (0.31) | 34.03 |
| | 2018 | Kougarok | 4655 / 96 (31) | 24.20 (0.88) | 0.30 (0.21) | 75.23 (0.57) | 23.15 |
| | 2019 | Teller | 5376 / 199 (69) | 38.21 (0.36) | 0.38 (0.09) | 106.74 (0.38) | 39.75 |
| **Total** | | | 23481 / 653 (203) | | | | |





**Table 3. Comparison of model performance for linear regression, GAM, and random forest models that are trained on individual years and individual sites.**


| Study Site | Year | Model | Train R² | Test R² | Train RMSE | Test RMSE |
|---|---|---|---|---|---|---|
| **Teller** | **2017** | Linear Regression | 0.34 | 0.31 | 7.23 | 7.34 |
| | | GAM | 0.50 | 0.45 | 6.28 | 6.56 |
| | | Random Forest | 0.96 | 0.77 | 1.66 | 4.28 |
| | **2018** | Linear Regression | 0.35 | 0.32 | 8.88 | 9.02 |
| | | GAM | 0.53 | 0.48 | 7.55 | 7.90 |
| | | Random Forest | 0.96 | 0.77 | 2.07 | 5.30 |
| | **2019** | Linear Regression | 0.29 | 0.37 | 13.69 | 12.74 |
| | | GAM | 0.50 | 0.52 | 11.45 | 11.07 |
| | | Random Forest | 0.96 | 0.72 | 3.40 | 8.51 |
| **Kougarok** | **2018** | Linear Regression | 0.45 | 0.44 | 12.02 | 13.60 |
| | | GAM | 0.70 | 0.72 | 8.96 | 9.68 |
| | | Random Forest | 0.98 | 0.92 | 2.42 | 5.20 |

**Table 4. Random forest results for the training and testing data used to estimate SWE.**

| Study Sites | Year | Train R² | Test R² | Train RMSE (cm) | Test RMSE (cm) |
|---|---|---|---|---|---|
| Teller & Kougarok | All | 0.98 | 0.86 | 2.41 | 5.85 |
| Teller | All | 0.97 | 0.83 | 2.58 | 6.25 |
| | 2017 | 0.96 | 0.77 | 1.66 | 4.28 |
| | 2018 | 0.96 | 0.77 | 2.07 | 5.30 |
| | 2019 | 0.96 | 0.72 | 3.40 | 8.51 |
| Kougarok | 2018 | 0.98 | 0.92 | 2.42 | 5.20 |


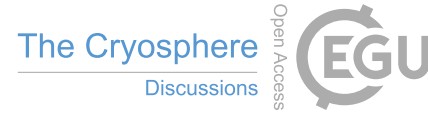

**Appendix**





Figures

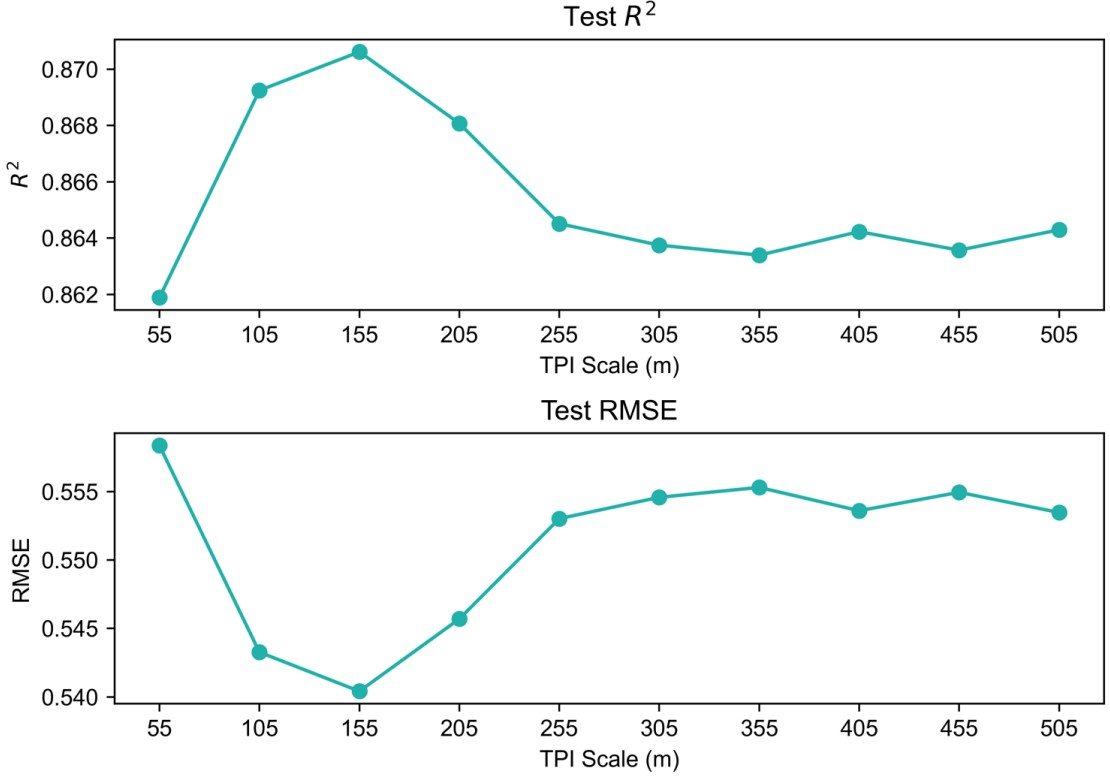

Figure A1. Random forest model performance for model runs using a range of TPI scales. The model performance is optimized when the TPI scale is 155 m.




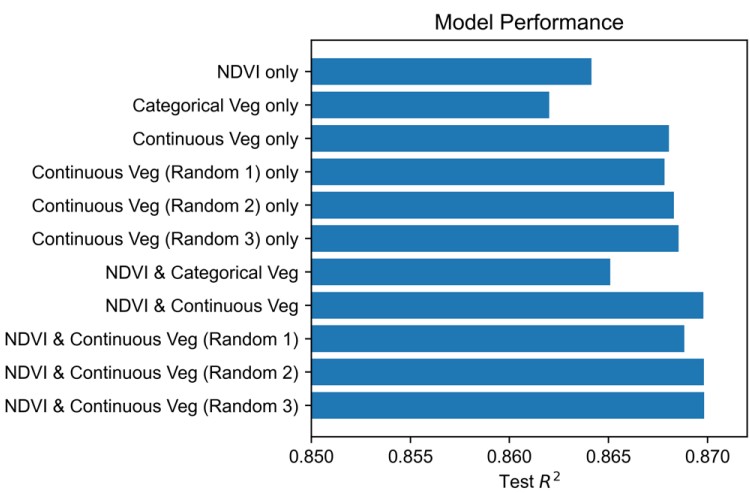


Figure A2. Random forest model performance for model runs using various combinations of vegetation features. The categorical vegetation uses the vegetation types shown in Figure 3. The continuous vegetation is a ranking ordered by which vegetation type has higher SWE. The continuous vegetation ranking is also randomized three different ways to determine if the ranking order is important.






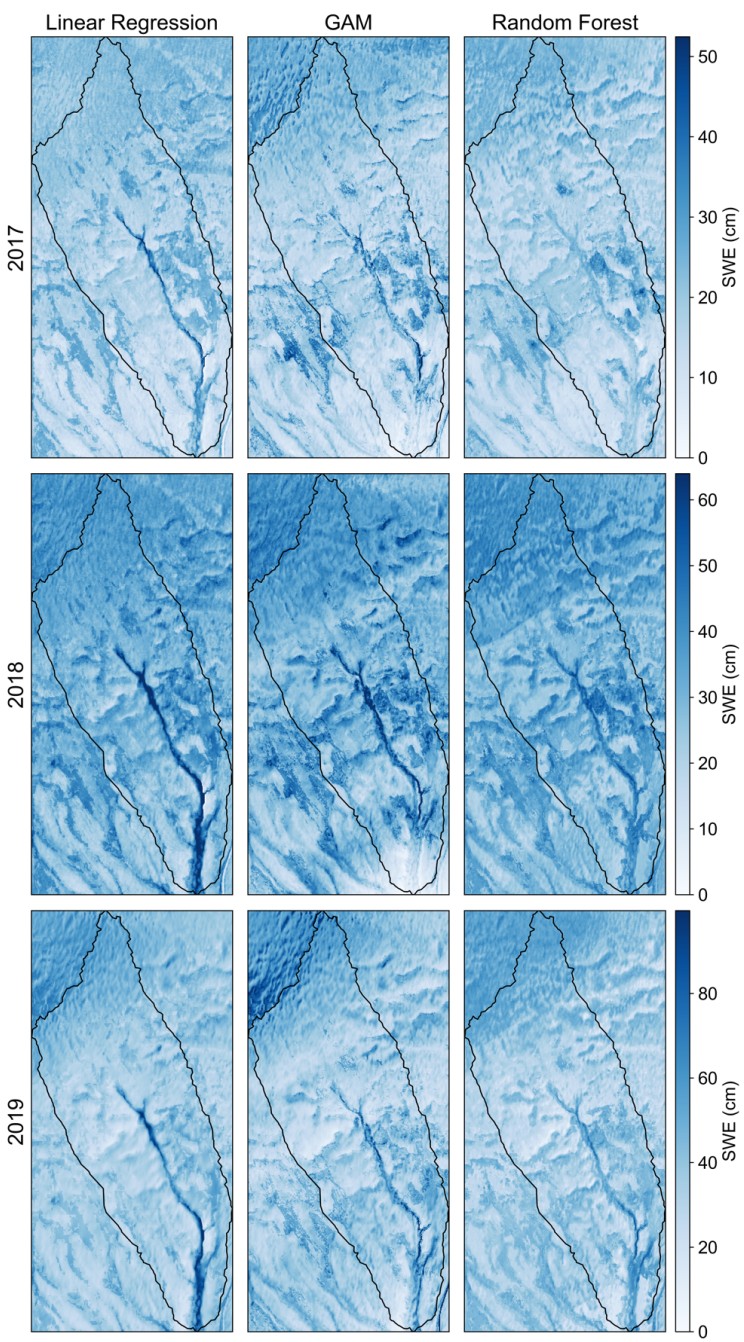

Figure A3. All three statistical models trained on individual years at Teller, without "year" as a feature. The
scale of SWE changes from year to year depending on the annual snowfall.



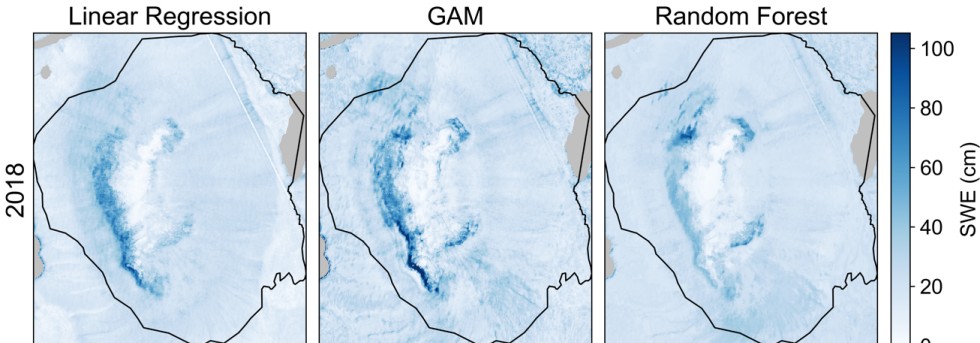

Figure A4. All three models trained on single year (2018) at Kougarok, without "year" as a feature.




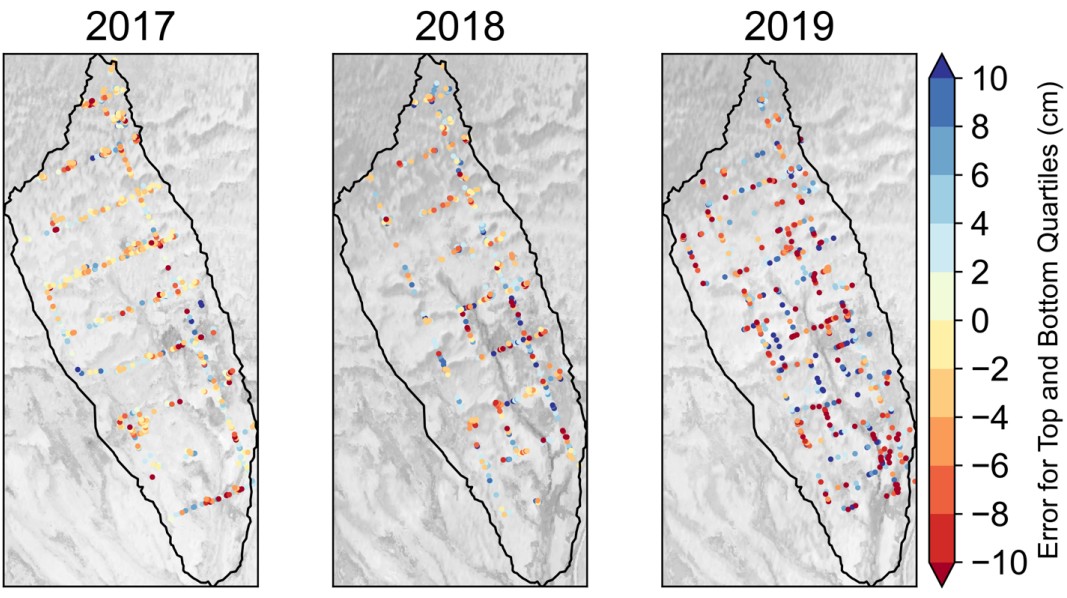


Figure A5. Spatial error (predicted minus observed) for the upper-most and lower-most quartiles of error in the random forest SWE prediction for the final model at Teller.






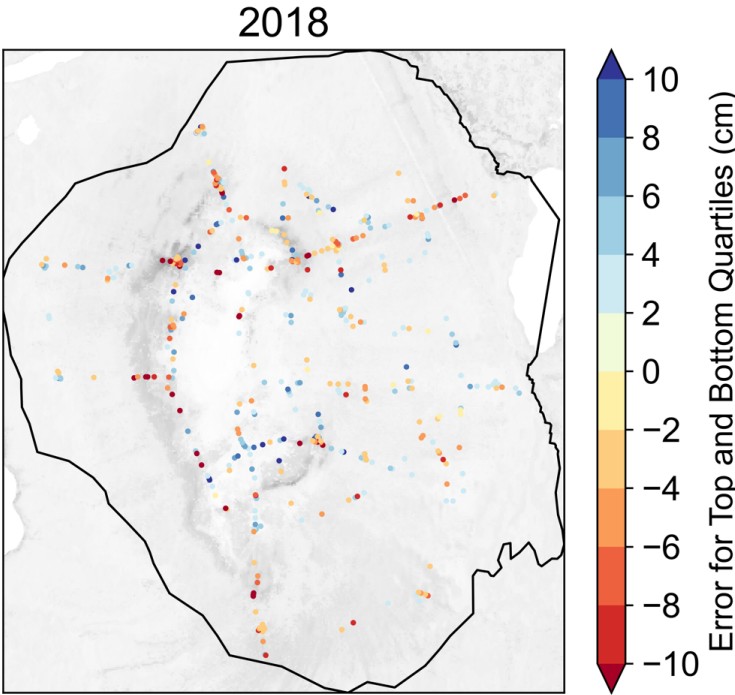

Figure A6. Spatial error (predicted minus observed) for the upper-most and lower-most quartiles of error in the random forest SWE prediction for the final model at Kougarok. The error is not distributed systematically through space.




## Tables

Table A1. Vegetation classes ranked by observed SWE from lowest SWE (1) to highest SWE (12).

| Vegetation Type | Rank |
|---|---:|
| Dryas-lichen dwarf shrub tundra | 1 |
| Birch-Ericaceous-lichen shrub tundra | 2 |
| Ericaceous dwarf shrub tundra | 3 |
| Sedge-willow-Dryas tundra | 4 |
| Willow-birch shrub | 5 |
| Alder-willow shrub | 6 |
| Tussock-lichen tundra | 7 |
| Wet meadow tundra | 8 |
| Wet sedge bog-meadow | 9 |
| Mesic graminoid-herb meadow tundra | 10 |
| Mixed shrub-sedge tussock tundra | 11 |
| Willow shrub | 12 |




Table A2. Average winter (October-March) wind speed and prevailing direction.

| | | Wind Speed | Wind Direction (Prevailing) | Wind Direction (Prevailing) > 5 m/s |
|---|---|---|---|---|
| | | (m/s) | (°) | (°) |
| **Nome (measured at 5 m height)** | 2016-2017 | 10.14 | 60.40 (ENE) | 65.16 (ENE) |
| | 2017-2018 | 12.42 | 66.11 (ENE) | 67.71 (ENE) |
| | 2018-2019 | 12.06 | 76.57 (ENE) | 78.27 (ENE) |
| **Teller (station at top of watershed)** | 2016-2017 | 4.99 | 83.63 (E) | 98.17 (E) |
| | 2017-2018 | 5.47 | 94.66 (E) | 115.38 (ESE) |
| | 2018-2019 | 4.86 | 76.10 (E) | 92.07 (E) |
| **Kougarok** | 2018-2019 | 6.74 | 60.23 (ENE) | 45.00 (NE) |


Table A3. Hyperparameters for Random Forest Model

| | Final Model | Teller | Kougarok |
|---|---|---|---|
| Training size | 0.8 | 0.7 | 0.8 |
| Tree density | 300 | 400 | 600 |
| Max depth | 25 | 20 | 17 |
| Max features | 4 | 2 | 4 |
| Min samples split | 3 | 2 | 2 |
| Min samples leaf | 1 | 1 | 1 |
