# Peer review of "Spatial Patterns of Snow Distribution in the sub-Arctic"

_The Cryosphere, 2021_

## Author Comment (AC1)

Bennett et al., perform a detailed experiment at a set of study sites in Alaska to answer the increasingly important question of "how much snow exists here". Their work examines a suite of regression models of varying sophistication to model SWE based on a set of environmental predictors like NDVI, elevation and wind. The paper was detailed, well written with a novel methodology and promising resulting model performance from the RF (suggesting followup work in this area of research). While portions of the paper are a bit verbose, after some edits I believe that this paper would be an important scientific contribution for the readers of The Cryosphere.

Response: thank you for your review and for your positive comments on our work.

**Major Comments/Revisions:**

1. While the paper by Bennet et al., is generally well written, it can be overly detailed in certain places. With some restructuring, I believe the paper can be much more concise and effective. For instance, the Introduction from lines 50-95 is likely unnecessary content. I would much rather get right into the meat of the problem at hand starting on line 96, as information about the importance of snow etc. can probably be a sentence or two with details left in references to previous literature. I have similar comments for Section 4 (specifically 4.1, 4.2, 4.3 and 4.4) which should not be in the results section and likely could be summarized in either the methodology or introduction in a paragraph or two. The beginning of 4.4 was answering questions I had about the model setup described in the methodology. These should absolutely be grouped together and the structure revised for clarity. Finally on this point, the discussion section 5 is again far too verbose and should likely be restructured with some of the details moved to the results section or moved to the Appendix. I would recommend limiting the discussion to a summary of uncertainties, sources of error and questions left unanswered from the results.

Response: To address the above comments, we have made the following changes to the paper:

- Removed lines 50-95 (four paragraphs from the introduction), starting with paragraph "Within the Arctic hydrologic cycle..." to final paragraph starting with "Many studies have"...
- We have replaced these paragraphs with a short paragraph referring to the importance of snow and references to previous work.
- For Section 4.0, we have not moved these paragraphs into the methods section. The methods section details how these tests were set up and then the results section details the results of all the testing and how we arrived at the final model configuration.
- We have gone through the text and shortened it up, reorganized the text, and focused the text in the Discussion section, and added a new section on Future Work.

2. Line 462, I am interested/potentially concerned about the extreme importance of Year on the accuracy of your RF. The authors mention that this Year variable is in some ways a proxy for

temperature/precipitation differences between years, and I would ask why not explicitly test for this? While I agree that this conclusion is probably correct, incorporating temperature and precipitation data from a well-validated reanalysis product like ERA5 or MERRA2 could help evaluate this hypothesis. It would also help explain which of these two variables is the most important. Furthermore, a Year variable really limits the robustness of this product for applications outside of your current study and removing it would help in predictions elsewhere. For instance, what if you want to apply your model to data retrieved last year? Would the RF understand the year value of 2020 if fed into the model? However, it could, in theory, incorporate precipitation/temperature data from 2020 without issue.

Response: To test this we ran several new tests to understand the impact of year in our analysis. We ran the following tests within the random forest modeling framework: a) using Year resorted (3 tests), b) using Daymet precipitation for Year (1 test), c) using Nome Airport precipitation for impurity (1 test), and d) using ERA5 precipitation for Year (1 test), e) using ERA5 temperature for Year (1 test), and f) using ERA5 precipitation and temperature for Year (1 test). When we tested precipitation and temperature, we used total winter precipitation and average winter temperature (Oct - Mar) for each of the years. In Table R1-1 below we show all the prediction and importance metrics for each test.

As you can see, the overall fits (R2 and RMSE values for actual SWE vs predicted SWE) for each test did not vary that much across all of the tests. Factor importance did change slightly, with the overall four top factors remaining the same in almost every case except for ERA5 precipitation and temperature. Year and NDVI swapped in importance for the third test for impurity, with NDVI becoming the most important factor when 2018 and 2019 were interchanged (while permutation importance did not change).

We replaced Year with three different data sets, including Daymet (Thornton et al. 2020, Figure R1-1), Nome Airport climate station, and ERA5 (Muñoz, 2019). For each of these, we also show the histograms of the SWE prediction for train, test, and whole area for each year. As you can see, the histograms do not change much between Daymet and Nome Airport station as Daymet is likely using the Airport station to estimate precipitation values. ERA5 is also quite similar with a few differences. Factor importance does change between these, Precipitation and NDVI swapped in importance when using Daymet and Nome airport data in tests, with NDVI becoming the most important factor for impurity importance. For ERA5, precipitation was the most important factor for each importance metric.

We also did two tests using ERA5 temperature, and using ERA5 precipitation and temperature. We found these results were the most different in terms of changing our modeling predictions, and also factor importances. In particular, for ERA5 precipitation and temperature tests, the top four important factors were NDVI, elevation, TPI, and Precipitation (elevation, NDVI, Year, TPI) for impurity (permutation), respectively, with temperature as either the third-to-last (impurity) or fifth (permutation) most important factor. Thus, we decided to use only precipitation as we feel that it is the most important factor and likely our Year factor represents precipitation over temperature in our study.

Due to the similarity in responses, but the wider availability of ERA5, and the fact that it does not rely on a particular station proximity, we decided to replace ERA5 precipitation in our final model and have rerun all of our analysis and output new results for the paper. We have adjusted the text of the paper as well to reflect this update. Using ERA5 will also allow us to generate the model for different years outside of our field period, and additionally, allow us to generate data for the entire Seward Peninsula region, which will be useful as we begin to compare these results and consider how to apply them within subgrid-scale earth system modeling.

---

## Author Comment (AC2)

Review Response The Cryosphere #2

Review of the manuscript "Spatial patterns of snow distribution for improved earth system modeling in the artic"

The research presented by Bennet et al., exploits a large dataset of manual snow observations in two sub-artic study areas to understand snow distribution (snow depth, snow density and snow water equivalent) with different statistical methods. The analyses they have applied are correct and rigorous and the results obtained would be of interest to the broad audience of this journal. Additionally the database they have generated is highly valuable for the community. Congratulations for such a big work (more than 23000 manual snow depth and 600 density acquisitions!)

Nonetheless I think the manuscript still needs further work, which I am sure the authors will be able to carry out. This way, I recommend the publication of this work after a major review.

Response: Thank you for your detailed and thoughtful comments on our paper. We have addressed each comment below and hope that our answers are helpful. We think the paper has improved considerably with your comments and review.

Below I provide a list of minor points that must be taken into account along the manuscript. These are my major concerns about the work:

- Maybe, the most interesting finding is the importance of the NDVI to explain SWE spatial distribution. However, the NDVI is an index obtained in a particular date (in July). The NDVI in late summer, early autumn might be very different. This point must be discussed, highlighting the importance (or not) of obtaining the maximum NDVI along the year. Please add references to justify NDVI evolution in sub-arctic areas with dominant presence of shrubs.

Response: For our NDVI analysis we did look at different periods of the summer and also we considered different NDVI scenes to understand how NDVI was shifting through the summer period and year to year. NDVI in late July and August in Alaska represents peak greenness (Boelmann et al. 2011). One issue with using NDVI and any type of remote sensing in the Arctic is that there is a lot of cloud cover, so finding a clear image to use can be challenging. However, we found several images with clear-sky NDVI available in the WorldView2 database (1-5 m) (Table R2-1).

We also examined the area over the Teller watershed only using Landsat imagery (30 m resolution) acquired via Google Earth Engine for a separate analysis where we considered NDVI changing over time, using dates from late June, late July, and August (after the 15 of the month, Figure R2-1). We see an increasing temporal trend in NDVI, or in other words, an increase in greenness as a measure of plant productivity over time and a difference in the monthly NDVI, with the highest values in late July and late August representing the maximum greenness.

[Figure]

Figure R2-1. Changes in the Teller water NDVI over time from Landsat imagery between 2000 and 2021. Each of the dots represents the NDVI calculated from one Landsat image during the growing season (defined as June, July, and August) between 2000 and 2021.

Then, using WorldView2 imagery (~2 m resolution), we tested each date within our Teller model to determine how the different dates affected the representation of observed SWE. We see in these results that NDVI in late July and mid-to-late August from different years produce similar results in terms of our model performance. Thus, we went with the best performing data based on the $R^2$ and RMSE results for train and test, which are the NDVI data from the dates presented in the paper. We have now added some discussion of this testing and results into the paper and also some references to support our discussion. Please see the track changed version of the paper.

Table R2-1. NDVI from different dates for the Teller watershed

| Dates | R2/RMSE^ Train Data | R2/RMSE Test Data |
|---|---|---|
| 2011-07-27* | 0.96/0.27 | 0.84/0.57 |
| 2016-07-31 | 0.96/0.27 | 0.83/0.58 |
| 2019-08-22 | 0.97/0.26 | 0.82/0.59 |
| Mean | 0.97/0.25 | 0.84/0.57 |

*Used in final analysis

^RMSE is in square root values

We would like to note that we also find these results really interesting, and thus we are writing a separate paper that will be focused mainly on examining the NDVI differences and findings within the work to both investigate what we are seeing and delve into it more deeply; thus, presenting this level of detail in the current work is considered outside of scope.

- ● Authors claim that the NDVI might describe shrubs presence ("NDVI in our study likely reflected the taller, denser shrubs patterns present in the landscape." Or "NDVI, that we believe represents shrub pattern ing"). I think this is interesting, but in the manuscript there is not any analysis to sustain this. Why not to correlate or analyze the distribution of vegetation types (already exploited in the manuscript) with the NDVI? This point must be tackled conveniently or all affirmation regarding NDVI -shrubs "relation" removed.

Response: To address this question, we compared NDVI used in our study to the vegetation classification map created using hyperspectral imagery (Konduri and Kumar, 2021) for the Teller watershed. We overlaid these on a lidar-derived DEM for resolution and found that the vegetation type with the highest NDVI is willow shrubs (Figure R2-2). Willow shrubs are the only vegetation type present at Teller with full-size shrubs (i.e., not dwarf shrubs in tundra vegetation types). Willow shrubs were also the only vegetation type with no statistical overlap with other vegetation types; the NDVI of willow shrubs is significantly different than all other vegetation types, with other vegetation types showing similarities between groups, indicated by the results of an ANOVA and Tukey's HSD test. Thus, we do believe that high NDVI values in our study represent taller, denser shrubs. We have decided to add this figure to the manuscript as Figure 5.

[Figure]

Figure R2-2. NDVI as it relates to vegetation type in the Teller watershed. Figure 5 in paper.

We think that NDVI as applied in our study is useful to use over the vegetation classification because NDVI data are widely available, while detailed vegetation data classification is not. We

recognized that the paper was missing a detailed discussion of the fact that NDVI and vegetation really gave us quite similar results in our analysis (Figure A4). We have now added this missing detail to the Results and Discussion sections of the paper.

- Methods section must be reorganized. It is too dense and in some cases, it is not easy to understand all analyses performed. For instance Model Implementation section is too general and in some paragraphs it explains random forest implementation, then came back to GAMs…please present it more organized. Moreover, some analyses applied in results section are not described in methods section (i.e. correlations between snow depth, snow density and SWE).

Response: We have worked to reorganize the Methods section and pare down the wording through the text. However, some of the results such as correlations between snow depth and density have not been moved because we believe that they are Results. The paper details a big effort, so we think moving all of those findings into Methods would simply bulk this section up. If this is somehow a sticking point for the editor and reviewer, we can revise it.

- The writing is sometimes too repetitive. Many sentences can be removed and some of them might be shortened without losing information. In some sections there are too vague affirmations ("we believe", "likely reflected "…), which are not sustained with the results. This type of sentences along the manuscript must be rephrased or removed. In this regard in several sections of the manuscript there are statements of ongoing work or future applications of the results obtained. Some of them are repetitive and can be removed or at least be all of them grouped in a new section in the discussion of "future work".

Response: We have gone through and done significant revisions and reorganized sections. We removed any wording such as 'we believe'. We have grouped all the statements regarding future work into a section by that name.

Minor comments:

Title: From my understanding the title is too wide. The manuscript does not analyze or work with earth system models. The research analyses different features that controls snow distribution (SWE, snow density and snow depth) with different approaches (GAMs, random forests…). Please change conveniently. Here some suggestions: "Understanding of snow spatial patters with statistical approaches in artic areas", "Snow spatial patters in artic areas analyzed with random forests"….

Response: Thank you. We have adjusted the title to remove the reference to earth system models.

Abstract: Authors state that both sites are sub-arctic (line 17). Why do you state that this is artic? I don't see any problem to state along the manuscript that you are characterizing snow distribution in sub-artic areas.

Response: Yes, we have now shifted all the wording in the abstract text and elsewhere (e.g., the title) in the manuscript to refer to the sub-Arctic and sometimes the Arctic where appropriate.

Sentence from line 27 to line 31 is useless. For sure this information will be used to improve other models and the understanding of hydrology, topography…but is not needed in the abstract as far it does not summarizes your study or the results you obtain. Remove it.

Response: We agree that this component of the Abstract is not about the results of the study, but it is noting the major driver of the study and the work as a whole, which is to improve the characterization of snow distribution and validate results within the E3SM earth system model. We have left this sentence in, but shortened/tightened it up.

Keyword: remove Machine learning and permafrost (it is interesting that permafrost is close or even present in the study areas but I don't see this as a key word), include random forests and change artic by sub-artic.

Response: We have made these changes as suggested.

Line 75: Include more recent references (and maybe remove the oldest). Even if most of those suggested below are mountain area studies, the information and the results these obtained are highly interesting in this topic. Moreover these references may be useful in the subsequent paragraph where no references are included after the first sentence.

- Mendoza, P. A., Shaw, T. E., McPhee, J., Musselman, K. N., Revuelto, J., & MacDonell, S. (2020). Spatial distribution and scaling properties of lidarâderived snow depth in the extratropical Andes. *Water Resources Research*, *56*(12), e2020WR028480.
- Mott, R., Vionnet, V., & Grünewald, T. (2018). The seasonal snow cover dynamics: review on wind-driven coupling processes. *Frontiers in Earth Science*, *6*, 197.
- Revuelto, J., López-Moreno, J. I., Azorin-Molina, C., & Vicente-Serrano, S. M. (2014). Topographic control of snowpack distribution in a small catchment in the central Spanish Pyrenees: intra-and inter-annual persistence. *The Cryosphere*, *8*(5), 1989-2006.
- Schirmer, M., Wirz, V., Clifton, A., & Lehning, M. (2011). Persistence in intraâannual snow depth distribution: 1. Measurements and topographic control. *Water Resources Research*, *47*(9).
- Trujillo, E., Ramírez, J. A., & Elder, K. J. (2007). Topographic, meteorologic, and canopy controls on the scaling characteristics of the spatial distribution of snow depth fields. *Water Resources Research*, *43*(7).
- Vionnet, V., Guyomarc'h, G., Bouvet, F. N., Martin, E., Durand, Y., Bellot, H., ... & Puglièse, P. (2013). Occurrence of blowing snow events at an alpine site over a 10-year period: Observations and modelling. *Advances in water resources*, *55*, 53-63.

Response: We have adjusted this section of the paper significantly to address Reviewer #1's comments. However, we have retained a section that refers to the overall importance of snow distribution modeling, where we have included most of these references, including Mendoza, Mott, Revuelto, Trujillo, and Vionnet. We have included the Schirmer paper later on in the

discussion section. Thank you for suggesting these articles, they were really interesting to read and we think their addition contributes to this section of the paper greatly.

Line 99-101, Also cite other models as, FSM, snowpack and crocus. Despite the article is mainly focus in artic (sub-artic) areas, it is continuously doing references to mountain area works (which from my understanding are needed in this research), so I consider it is worthy to cite these physically based models.

- Essery, R.: A factorial snowpack model (FSM 1.0), Geosci. Model Dev., 8, 3867–3876, https://doi.org/10.5194/gmd-8-3867-2015, 2015.
- Vionnet, V., Brun, E., Morin, S., Boone, A., Faroux, S., Moigne, P. L., ... & Willemet, J. M. (2012). The detailed snowpack scheme Crocus and its implementation in SURFEX v7. 2. *Geoscientific Model Development*, *5*(3), 773-791.
- Bartelt, P., & Lehning, M. (2002). A physical SNOWPACK model for the Swiss avalanche warning: Part I: numerical model. *Cold Regions Science and Technology*, *35*(3), 123-145.

Response: Thank you, we have added references to these models in this section of the paper.

Lines 134 to 140 are not needed in the introduction. Maybe you can include a brief reference to the validation you mention here in the discussion in a "future work" section. Nonetheless in the introduction it does not support the findings of this research.

Response: We have removed this sentence from the Introduction and created a Future Work section of the paper at the end before the conclusions. See below for other information from the paper that we have moved into this Future Work section.

Line 151. Which is the distance to the airport (in km) from both sites?

Response: Although the Nome airport station is far away from the study sites (35 km from Teller, and 78 km from Kougarok, approximately), there are consistencies, particularly in the temperature records, that make the Nome airport station information a good proxy for the research sites. For example, the minimum and maximum temperatures during the study period (shown for where records overlap) are quite similar (Figure R2-3). Our weather stations at the study sites does not have a longer-term climatological record available, thus the Nome airport station is referred to in this section.

[Figure]

Figure R2-3, minimum and maximum daily temperatures from 2017-2019 for Nome Airport, Teller and Kougarok stations.

Line 162 to 165: This information is not needed in this research. Remove it.

Response: We have removed this sentence and the details of surficial geology in the next paragraph detailing the Kougarok site.

Line 182: Why you didn't include the data from Teller in 2016? If you don't use it I think it is not necessary to provide this information here.

Response: Data from the initial year of snow measurements in Teller were not conducted using the same instrumentation and approaches applied in 2017-2019. We have removed this sentence.

Line 188. Add Snow-Hydro reference :

● Sturm, M., & Holmgren, J. (2018). An automatic snow depth probe for field validation campaigns. *Water Resources Research*, *54*(11), 9695-9701.

Response: We have added this citation.

Line 190: SWE coring tube was evaluated in this study:

● LópezâMoreno, J. I., Leppänen, L., Luks, B., Holko, L., Picard, G., SanmiguelâVallelado, A., ... & Marty, C. (2020). Intercomparison of measurements of bulk snow density and

water equivalent of snow cover with snow core samplers: Instrumental bias and variability induced by observers. *Hydrological Processes*, *34*(14), 3120-3133.

This work must be cited here as SWE coring tube was evaluated here.

Response: We have added this citation. Thank you.

Line 200: You are using an interpolated snow density value for each snow depth acquisition in order to determine the SWE for each location. I encourage manuscript authors to include a statement about the error introduced with such an approach.

Response: Yes, that is correct. We collected 96-234 density measurements each year, but thousands of snow depth data points (see Table 2). Thus, we had to interpolate the density in some way. While we tested some different approaches, and attempted to examine the length of autocorrelation within the density data, there is uncertainty associated with the approach. We have added a few words about this uncertainty to the end of the line. Please let us know if you would like more details on this.

Line 211 Include this reference in which TPI was presented.

- Weiss, A. (2001). Topographic Positions and Landforms Analysis (Conference Poster). ESRI International User Conference. San Diego, CA, pp. 9-13.

Response: We have added this reference.

Paragraphs from line 210-215 and 216-225: these paragraphs can be reordered and combined. When TPI surrounding square window are defined is a bit hard to understand why distances between 15 m and 155 m are selected. In the next paragraph this is explained, but it is hard to understand lines 213 to 215 as now are stated. Maybe you can change the order of these two paragraphs.

Response: We have reorganized these two paragraphs, condensed them into a single paragraph, and adjusted some of the text to improve our explanation.

Line 231: here it is stated "…vegetation type was included in the model as…". At this point of the manuscript models have not been described. I would rephrase this sentence removing the reference to "the model".

Response: We have rephrased this sentence.

Is the vegetation ranked with yearly SWE values and then this information used as a feature in the models? If yes, I think this is not conveniently done. Why don't you directly use the map from Konduri and Kumar, 2021?

Response: We considered vegetation as both a categorical and continuous variable in our analysis (see Appendix Figure A2, and discussion of this in section on Modeling). We tested

numerous different ways of using the vegetation data from Konduri and Kumar, 2021 in both continuous and categorical (i.e., using the data directly) and found that our categorical ranking resulted in lower random forest model $R^2$ values. We tested different ways of ranking the vegetation data as well. Actually, ranking the data randomly and ranking using the SWE data gave us almost the same results. However, it did not make sense to us to rank the vegetation data randomly. So, in the absence of having a better method, we used the SWE data.

Line 247: Add these references:

- Winstral, A., Elder, K., & Davis, R. E. (2002). Spatial snow modeling of wind-redistributed snow using terrain-based parameters. *Journal of hydrometeorology*, *3*(5), 524-538.
- Mott, R., Schirmer, M., & Lehning, M. (2011). Scaling properties of wind and snow depth distribution in an Alpine catchment. *Journal of Geophysical Research: Atmospheres*, *116*(D6).

Response: We have added these references.

Line 257-265: "W" is West aspect or the Wind factor, please clarify. The method used to define aspect factor is not clear. It seems that different works have already followed this method (Dvornikov et al., 2015; Evans et al., 1989; Liston and Sturm, 1998), however this manuscript may benefit of a more accurate description about this method and how it is applied.

Response: We have added an 'f' to make it Wf for wind factor to differentiate it from the W used to abbreviate the cardinal direction West. Although these other papers describe a similar use of a wind factor, we synthesized several different methods to apply this simple wind and aspect estimate based on the equations we illustrate in the paper. We have added some more detail to the Appendix to describe this in more detail as we do think a clear explanation and detail is missing from existing literature.

Random forests sections. Here you must cite:

- Liaw, A., & Wiener, M. (2002). Classification and regression by randomForest. R New s, 2,6.

Moreover I encourage manuscript authors to provide more details about random forests and how these works with an appropriate language (for example I don't see appropriate to talk about "votes").

Response: We now cite the Liaw and Wiener 2002 publication. Votes is used in the suggested reference to describe random forest prediction and this terminology, in general, is commonly used in descriptions of random forest models. We have left this in the text. We have not added to the description of random forest modeling because we feel our detail is appropriate.

Line 309: Is there any reason to "square root " the SWE? Please explain why and include references/arguments to justify this decision

Response: We use a square root transformation of SWE to ensure the distribution is normal. This is necessary for the linear and GAM statistical modeling, although not for the random forest. We have added a few words to say this in the text. Transforms such as this are applied widely in modeling approaches, and we do not feel we need a reference or argument to justify its use.

Line 310; include here the split sample %.

Response: We have added the split sample % to Section 3.1.4.

Line 313-319: The information detailed in this paragraph about all subsets used can easily be summarized in a table. Please do it and give names/acronyms to these subset models to then use them in results section.

Response: We considered this but the table would be very basic, which we do not think would add anything to the paper. Furthermore, adding acronyms to the models (i.e., calling it T17 or something like that) would add confusion for the reader. As we describe, we use precipitation (in our updated model) for the final model, and then the individual years do not include precipitation (since it would only be one value). All the other tests we performed (100s of them) are described separately in a limited way to not overwhelm the reader (i.e., testing NDVI vs vegetation). We do not think it would make sense to introduce all of those tests and describe them since we think this would add confusion and unnecessary detail to the paper.

Line 321: In line 303 authors state: "feature importance can be difficult to interpret in comparison to linear modeling or GAM approaches" and in this line it is stated "random forest performed the best of the three models and has the most comprehensive feature importance metrics". These two sentences are contradictory, please change conveniently and be consistent with the statements included in the manuscript.

Response: We intended for the first statement, "feature importance can be difficult to interpret in comparison to linear modeling or GAM approaches," to mean that the simplicity of linear and GAM models allows for easily interpretable feature sensitivity (e.g., for linear models, a given increase in x results in a given increase in y). Random forests do not have a similar way to explain how the feature importance metrics impact the target variable. However, we omitted this statement in the paper to avoid confusion and since it is not essential to the focus of the paper. We left the second statement as-is because random forests do have more comprehensive feature importance metrics compared to linear and GAM models.

Line 326: "we measured the value of each input feature in predicting" you mean feature importance or contribution to the model? One may expect that value is referred to the variable values (NDVI, value, TPI value….). Please be consistent along the manuscript with this definition of features contribution to models prediction capabilities.

Response: We removed the word value and replaced it with contribution, and checked for any other inconsistencies in the paper.

Line 329: Include this reference here:

- Louppe, G., Wehenkel, L., Sutera, A., & Geurts, P. (2013). Understanding variable importances in forests of randomized trees. In C. J. C. Burges, L. Bottou, M. Welling, Z. Ghahramani, & K. Q. Weinberger (Eds.), Advances in neural information processing systems 26 (pp. 431–439). Inc.: Curran Associate

Response: We have included the reference.

Line 337: Add: both importance metrics (MDI and MDA)-

Response: We added this to the text.

Line 340 to 342: This sentence must be moved before describing all models (line 308), as variance inflation and correlation coefficients are computed previously in order to analyze collineraity before applying the models.

Response: We have now moved the sentence up in the text as suggested.

Line 361 to 364: This sentence must be removed; this information is already provided in section 2.2

Response: We removed this sentence and transferred the information on snow depth sampling spacing to Section 2.2.

Line 386: Where are observed the solifluction lobes. Please show it in Figure 4 or remove this sentence.

Response: We clarified the presence and discussion of topographic features in the paper. We now discuss the presence of terraces and risers and show an example of their location in Figure 1 and Figure 4. And in Figure 4, their impact on TPI shown as alternating red and blue is mentioned in the caption.

Line 389 to 392. Where is this information (wind and aspect) shown? It is shown in Figure 4, so please include it also here.

Response: We are not completely sure what the reviewer means here, but we added (Figure 4a, b) to the text after this sentence.

Line 402:  How was determined the 20-80% split to train/validate the model?  Why 300 trees? Please include some references of previous works or explain why you chose those values

Response: We have changed the hyperparameter selection process to the Bayesian Search method following Reviewer #1's suggestion, and we have added more details in Section 3.1.4 about selection of hyperparameters with Bayesian Search to make this clearer. We chose the 20-80% split because the model performance does not change significantly when the training set is increased beyond 80%.

Line 405: Several tests with same configuration except TPI distance? In which test site? With all years?? Please clarify.

Response: We have clarified that we ran the optimization tests on each site and all years.

Line 409: NDVI model tests. Same questions of previous comment must also be answered here.

Response: We have clarified that we ran the optimization tests on each site and all years.

Line 426: In figures A3 and A4 are shown the SWE distribution maps obtained with the different models but here it is not clarified the % of the data used for validation and for training. Are you using all data for training and then you plot model results? This point must be clarified.

Response: These maps show the predicted SWE based on the final models developed using observed SWE and input factors using the split sampling approach. The models are trained on 80% of the data, which is explained in the methods section.

Line 430: Change this title. You are also doing a prediction of SWE in previous section.

Response: We changed the title of the previous section and moved some of the text around to better reflect the modeling optimization and testing results vs the SWE prediction results.

Line 444: Figure 7 SWE maps are obtained with the model that included data from all years and sites. Clarify.

Response: We developed the model using all years and all sites. We are unsure how to be clearer here.

Line 447 Authors know where stream saws are or, where the permafrost slump is located. Oppositely readers, who do not know these study sites, are not familiarized with these landforms. Show it in figure 7 (mark it with lines, an arrow…).

Response: We have added references and notation to these features in Figure 1 that illustrates the study sites.

Line 455: Is there any reason to justify why higher SWE have higher errors? This point must be discussed later.

Response: The 2019 snow survey was more spatially extensive and captured more variability in the SWE in this year. Deeper snow also might be more error prone due to the challenges with capturing snow depths beyond the limit of the magnaprobe. We do not know if the higher SWE is why we had more errors, but the data suggests this because the 2017 and 2018 surveys were most similar in terms of extent. We have adjusted this section of the text and added a Figure to the Appendix that shows the spatial extents of all the years of survey for Teller only.

Line 462: The sentence "even though year is ranked…of our study." From my point of view this sentence must be moved to the discussion. If the discussion includes a new section in which future work is explained

Response: We have moved this sentence to the Discussion section.

Section 4.7, SWE correlation between years. I really like heatmaps. However I don't see why authors didn't include correlation coefficients (pearson, kendall…) This will help to understand SWE correlations. Please compute correlation coefficients.

Response: We have computed the correlation coefficients and include them in the figure and text.

Lines 473 to 480 are redundant. This must be detailed in the introduction, not here.

Response: We have moved these lines and incorporated them into the introduction.

Line 503-504. I have found very interesting previous lines discussing the correlation of snow depth and density. However, I have not found in the manuscript any reference to the "no relationship for shallow snow (<60cm)". In figure 3b are shown all snow depth value. Moreover there are no previous references to the "60 cm" threshold. This result must be highlighted in results section. I encourage manuscript authors to add a new graph in figure 3 but including only snow depth values above 60 cm to show the positive linear correlation between snow depth and density.

Response: We have now updated the figure to show the linear regression for density using the results greater than 60 cm and added correlation values to the plot.

Line 512-516. As previously highlighted, I would appreciate to include here references to mountain area works, where the inter-annual consistency between SWE (and snow depth values) has also been observed.

Response:  We have added the studies below that have found inter-annual consistency in SWE distributions in mountain environments.

- Deems, J.S., Fassnacht, S.R. and Elder, K.J., 2008. Interannual consistency in fractal snow depth patterns at two Colorado mountain sites. *Journal of Hydrometeorology*, *9*(5), pp.977-988.
- Dozier, J., Bair, E.H. and Davis, R.E., 2016. Estimating the spatial distribution of snow water equivalent in the world's mountains. *Wiley Interdisciplinary Reviews: Water*, *3*(3), pp.461-474.
- Erickson, T.A., Williams, M.W. and Winstral, A., 2005. Persistence of topographic controls on the spatial distribution of snow in rugged mountain terrain, Colorado, United States. *Water Resources Research*, *41*(4).

- Winstral, A. and Marks, D., 2014. Long-term snow distribution observations in a mountain catchment: Assessing variability, time stability, and the representativeness of an index site. *Water Resources Research*, *50*(1), pp.293-305.

Lines 520 to 524: These lines must be included in a new section named "future work" (or similar).

Response: We have created a new Future Work section and added this text to this new section as suggested.

Line 535 to 541: This is also future work. Moreover some parts of these lines are redundant with previous statements of the work and can be removed.

Response: We have created a new Future Work section and added this text to this new section as suggested.

Line 543-545: Remove, already explained in results section.

Response: We have removed this text.

Line 548: Add references to justify that "consistent with those in previous studies in terms of how those factors affected snow distribution in the".

Response:  We added the following references to this sentence.

- Sturm and Wagner, 2010 (already cited in paper)
- Sturm et al., 2001a (already cited in paper)
- Sturm et al. 2001b (already cited in paper)
- Sturm, M., Douglas, T., Racine, C. and Liston, G.E., 2005. Changing snow and shrub conditions affect albedo with global implications. *Journal of Geophysical Research: Biogeosciences*, *110*(G1).
- Dozier, J., Bair, E.H. and Davis, R.E., 2016. Estimating the spatial distribution of snow water equivalent in the world's mountains. *Wiley Interdisciplinary Reviews: Water*, *3*(3), pp.461-474.
- Homan and Kane, 2015 (already in paper)
- Gisnås, K., Westermann, S., Schuler, T.V., Litherland, T., Isaksen, K., Boike, J. and Etzelmüller, B., 2014. A statistical approach to represent small-scale variability of permafrost temperatures due to snow cover. *The Cryosphere*, *8*(6), pp.2063-2074.
- Grünberg, I., Wilcox, E.J., Zwieback, S., Marsh, P. and Boike, J., 2020. Linking tundra vegetation, snow, soil temperature, and permafrost. *Biogeosciences*, *17*(16), pp.4261-4279.

Line 549; Landforms are not features included in the models. This sentence must be rephrased in order to highlight that "stream bed, permafrost thaw slump edges" tend to accumulate more snow.

Response: We have adjusted this sentence accordingly.

Line 551: NDVI values were obtained at a particular period of the year (in July). NDVI in late summer, early autumn might be very different. This point must be discussed, highlighting the importance (or not) of obtaining the maximum NDVI along the year. Please add references to justify NDVI evolution in sub-arctic areas with dominant presence of shrubs.

Response: We have added discussion on this point and added references. Please also see more details above response to this question in the Major comments section above.

Line 560-564: This sentence is too long and redundant. Please split it and remove unnecessary statements"

Response: We have reworded this section of the text.

Line 565: 300 m is a gradient in altitude? If yes, please state it, otherwise claify.

Response: We have clarified this in the text. Teller's maximum elevation is 300 m, and minimum is about 50 m.

Line 568-570: move to Future work section,

Response: We have moved this section to Future Work.

Line 572-574: Maybe show the correlation between TPI and NDVI in these study areas is interesting and shows. Just a suggestion.

Response: We are not completely sure of what the reviewer is asking. We feel that detailed study of NDVI with TPI relationships in this paper is outside the scope of the work. Thus, we have not added this to the current paper, but it will be the focus of future analysis and study, as noted in the Future Work section of the paper.

Line 574-575: Add references to justify that moisture accumulates here and this is associated with higher ecological productivity.

Response: We have added a reference to this sentence.

Line 580-581: Sentence for future work section.

Response: We have moved this to the Future Work section.

Line 590: UAS are drone observations?, These devices are usually named as UAV:

- Adams, M. S., Bühler, Y., & Fromm, R. (2018). Multitemporal accuracy and precision assessment of unmanned aerial system photogrammetry for slope-scale snow depth maps in alpine terrain. *Pure and Applied Geophysics*, 175, 3303– 3324.

- Harder, P., Pomeroy, J. W., & Helgason, W. D. (2020). Improving sub-canopy snow depth mapping with unmanned aerial vehicles: lidar versus structure-from-motion techniques. *The Cryosphere*, *14*(6), 1919-1935.
- Revuelto, J., LópezâMoreno, J. I., & AlonsoâGonzález, E. (2021). Light and shadow in mapping alpine snowpack with unmanned aerial vehicles in the absence of ground control points. *Water Resources Research*, *57*(6), e2020WR028980.

Nowadays UAV acquisitions of snow depth are accurate and very dense in space I would not cite López-Moreno et al., 2009, as the interpolation methods they described and evaluated is not suitable when working with UAVs. In the contrary I would cite the articles referred above.

Response: We have added these references to the section and replaced UAS with UAV.

Line 595: Remove this part of the sentence (which is not needed in the conclusion): "which is being undertaken in current work by the authors."

Response: We moved this part of the sentence to the Future work section.

Line 599-600: This affirmation has not been demonstrated in this study and must be removed: "this model may be used to estimate snow distribution beyond the study sites, work that is also ongoing by the authors".

Response: We moved this sentence to the Future Work section.

Line 600-603: This is future work, not a conclusion; I would remove it, or at least shorten it.

Response: We moved this sentence to the Future Work section.

Line 610: "that we believe represents shrub pattern". Believe something is not a conclusion. The results have shown that the NDVI is the most important feature to explain SWE with random forests, this is the conclusion. Similarly, TPI is a very important feature. What TPI represents is not an output of your research ("an index that represented the features in the landscape such as the stream bed and various topographic features including solifluction lobes" is an appreciation of the authors). Please change conveniently this section of the conclusions.

Response: We have edited this section of the text.

Line 611-615. Somehow this is future work. I would remove this sentence or move it to the discussion.

Response: We have moved this up to the Future Work section.

Conclusions section: I don´t see an interesting outcome of this research: the linear relation between snow depth and snow density for snow depth values above 60 cm. I would include it here.

Response: The finding has been added to the Conclusions.

Figures and tables:

Figure 2: In 2017, 2018 and 2019, was measured snow depth and density in same locations? If now, I encourage showing three separate maps with true locations for each year.

Response: No, it was not measured in the same locations exactly. However, we have opted not to change this figure and have added maps into the Appendix that show the same figure but different spatial arrangement of the data collected in each year for Teller.

Figure 3 See comments of lines 503-504

Response: See our response to the previous comment.

Table 1. Several distances to compute the TPI are used in the manuscript. I would remove the 155 m distance of TPI here and state that several distances were tested.

Response: The TPI value used as model input is 155 m, so we have left the table the same.

Figure 9: change graphs background to white. This will help to interpret the light yellow areas showing lower points density.

Response: We have revised the figure accordingly.

Figure A3, The fact that you did not include year as a feature in the model must be explained in the text not in the caption.

Response: We have adjusted this in the figure caption.

Figure A4 and A5, it is a bit hard to get an idea of error spatial distribution. You can also include a frequency histogram (small panel inside this figure) to provide a better overview of erros.

Response: Our point with the figure is that the error is distributed evenly in space across the study sites (there is no clear pattern of error). We think that the current maps do show this well enough.

**References**
Boelman, N. T., Gough, L., McLaren, J. R., and Greaves, H.: Does NDVI reflect variation in the structural attributes associated with increasing shrub dominance in arctic tundra?, Environ. Res. Lett., 6, 1–12, 2011.

---

## Author Response (AR1)

April 6th, 2022

Dear Cryosphere,

We have revised our manuscript and uploaded the new versions with changes. We have also added reference to Meloche et al. 2022 in to the text and references.

Thank you very much.

Katrina Bennett

---

## Author Response (AR2)

Dear Dr. Bennett and colleagues,

Thank you for the changes to your manuscript. The reviewers were supportive of the paper and its methodology, so I am more than happy to iterate the changes with you at this stage. More detail is needed in a few of the amendments to the manuscript as outlined below, and there are a few typos to correct:

Line 129. Remove distance to Nome airport: it's already stated in line 113 and the value is rounded up / inconsistent between the two lines.
Response: this has been removed.

Line 161 states there is uncertainty in the density. Please describe the methodology used to assess this, and the value of the uncertainty estimate.
Response: We did not quantify the uncertainty that was added to our analysis when estimating the density at each of the snow depth points. That would require data about the density distributions that we did not collect. Because of this, we wanted to recognize that using IDW to estimate density at each of the snow depth points adds some amount of uncertainty to the analysis without explicitly giving an error value for the uncertainty. However, we can characterize the variability of the observed density measurements, using the range and standard deviation values of the density dataset. We have added the variability metrics to the manuscript, and we added some citations to the original IDW interpolation methods articles, see the new sentences added to the track changed version of the manuscript and new references (below).

References (added to manuscript):

Franke, R., 1982. Scattered data interpolation: tests of some methods. *Mathematics of computation*, *38*(157), pp.181-200.

Zimmerman, D., Pavlik, C., Ruggles, A. and Armstrong, M.P., 1999. An experimental comparison of ordinary and universal kriging and inverse distance weighting. *Mathematical Geology*, *31*(4), pp.375-390.

Figure 5 / section 2.3.2 / line 202 'We assessed vegetation types' needs a methodological description. The response to reviewer 1 is fine: it just needs to be added to the paper.
Response: We edited the word assessed to 'binned' in the sentence and altered and edited the sentence. See track changed version of the document.

Line 364 suggests Figure 5 shows the results of the ANOVA and Tukey test, but does not. These results need to be added (probably in Table A1)….Line 205 suggests ANOVA and Tukey's test statistics are included in Table A1, but appear to be missing.
Response: We have adjusted the text in the manuscript in this section.

Line 420. It's not clear to me why greater variability and error can be expected with more measurements here as this will depend on the semivariogram. Have you plotted this for the study

years / sites? Were the other years undersampled? I would say that the spatial extent of 2019 is similar to the other years (fig A3) but the spatial resolution is higher.

Response: We have changed spatial extent to spatial resolution and edited these sentences to read as follows:

*Errors are higher in the years where there was higher SWE in the basin, such as in 2018 compared to the lower SWE year of 2017 in the Teller watershed, when the survey resolution was similar (Figure A3). In 2019, the survey captured a finer spatial resolution and thus we expect greater spatial variability and higher error.*

Line 524. Typo: Melosche -> Meloche. Please could you comment on why vegetation is of high importance in this study but low importance in Meloche et al?

Response: We corrected this topographic error in the text. In response to your second question, we think that this is also our finding with regards to NDVI as being an important variable to snow. We talk about this the opening paragraph to Section 5.3, what NDVI represented, and how they were similar in response in our work when we compared them directly in our model. We also discuss what we think NVDI represents (taller shrubs). We do intend to look deeply at NDVI and what it represents in another study that is being undertaken with data we collected in 2022. However, the paragraph of Section 5.3 summarizes all of this.

Figure 6 and 9 typo: Microtpopgraphy -> Microtopography

Response: Corrected. Thank you for catching this!

Description of wind factor: line 040 refers to figure A1, but should be A2.

Response: Corrected.

Figure A7, A8 are not referred to in the text but seem really interesting. Please describe in the text or remove if they add no information.

Response: We correct the references to these figures, and others, that were not updated when we added several figures to the Appendix in our last edit. We checked all Figures references through the text as well.

Many thanks and with best wishes,
Mel

Thank you.

Sincerely,

Katrina and co-authors

---

## Author Response (AR3)

Dear Dr. Bennett,

Thank you for the revised manuscript. The majority of points have been adequately addressed. Thank you for the detailed review! We really appreciate your care and attention to this paper.

However, three points are still outstanding:

- 'In 2019, the survey captured a finer spatial resolution and thus we expect greater spatial variability and higher error'. The coefficient of variation in Table 2 is lower for 2019 than the other years, so this does not suggest greater spatial variability with finer resolution. Is there any other evidence to indicate undersampling in other years? Once the spatial variability has been sufficiently captured then additional observations will not provide any more information. Could the higher error simply be a function of the higher SWE (e.g. percentage errors vs absolute errors).

Response: we checked all of the values and found that the CV reported in Table 2 has very small errors, thus the 2019 CV is in fact slightly higher than in 2018. However, the CV values for 2018 and 2019 are really quite similar (only one point apart previously, and are actually the same when you consider the entire SWE data set rather than the averaged spatial SWE values), suggesting about the same amount of spatial variability in these two years. In 2017 versus these two high years, the CV is much higher, and Kougarok 2018 also has a much higher CV for SWE (with lower SWE values). So, I don't think it can be simply a function of higher SWE. I suspect that at some point, there is a lot of variability in the snow pack, and as snow fills in the topography, it starts to be reduced, but all of this is likely influenced by things like sampling density, snow depths/SWE amount, and variations in measurement techniques/teams year to year. I have adjusted the text in this part of the ms, and noted that it is beyond the scope of the study to consider this point in more depth. I think we would need to set up a specific test to look at this, which would be interesting and something to think about for next year's survey. Note, in this checking, I adjusted a few of the values in the entire Table 2 slightly.

The response to 'Please could you comment on why vegetation is of high importance in this study but low importance in Meloche et al?' contains only information on why vegetation is important in this study. Please discuss why the two papers with similar methodology have very different vegetation importance.

Response: We have written a new paragraph that describes the Meloche et al. 2022 findings, compares it to our work, and notes why we feel our work is an improvement over their approach. Please see the track changed version of the text.

Please revisit the figure number referencing as these are still not fixed. Figure A8 is not referenced and there are other errors with the numbering.

Response: we fixed these typographic errors.

Many thanks and with best wishes,
Mel